# The anti-GD2 monoclonal antibody naxitamab plus GM-CSF for relapsed or refractory high-risk neuroblastoma: a phase 2 clinical trial

Jaume Mora [1] ✉, Godfrey C. F. Chan[2,3], Daniel A. Morgenstern [4], Loredana Amoroso[5,6], Karsten Nysom [7], Jörg Faber[8], Arthur Wingerter[8], Melissa K. Bear[9], Alba Rubio-San-Simon[10], Blanca Martínez de Las Heras[11], Karen Tornøe[12], Maria Düring[12] & Brian H. Kushner[13]

In this single-arm, non-randomized, phase 2 trial (NCT03363373), 74 patients with relapsed/refractory high-risk neuroblastoma and residual disease in bone/bone marrow (BM) received naxitamab on Days 1, 3, and 5 (3 mg/kg/day) with granulocyte-macrophage colony-stimulating factor (Days -4 to 5) every 4 weeks, until complete response (CR) or partial response (PR) followed by 5 additional cycles every 4 weeks. Primary endpoint in the prespecified interim analysis was overall response (2017 International Neuroblastoma Response Criteria). Among 26 responders (CR + PR) in the efficacy population (*N* = 52), 58% had refractory disease, and 42% had relapsed disease. Overall response rate (ORR) was 50% (95% CI: 36-64%), and CR and PR were observed in 38% and 12%, respectively. With the 95% CI lower limit for ORR exceeding 20%, the primary endpoint of overall response was met. Patients with evaluable bone disease had a 58% (29/50) bone compartment response (CR, 40%; PR, 18%). BM compartment response was 74% (17/23; CR, 74%). One-year overall survival and progression-free survival (secondary endpoints) were 93% (95% CI: 80-98%) and 35% (95% CI: 16-54%), respectively. Naxitamab-related Grade 3 adverse events included hypotension (58%) and pain (54%). Overall, naxitamab demonstrated clinically meaningful efficacy with manageable safety in patients with residual neuroblastoma in bone/BM.

Neuroblastoma is an aggressive childhood cancer that accounts for ~15% of all cancer-related deaths in children[1]. The average age of patients at neuroblastoma diagnosis is 1 to 2 years[2]. Approximately 50% of patients with neuroblastoma present with high-risk (HR) disease[2–5], most frequently with metastases in the bone marrow (BM; 70–89%) and bone (56–65%)[6–11].

High-risk neuroblastoma is a complex disease characterized by heterogeneous tumors and patients as well as differential responses

[1]Pediatric Cancer Center Barcelona, Hospital Sant Joan de Déu, Barcelona, Spain. [2]Queen Mary Hospital & Hong Kong Children's Hospital, Pok Fu Lam, Hong Kong. [3]The University of Hong Kong, Pok Fu Lam, Hong Kong. [4]The Hospital for Sick Children, University of Toronto, Toronto, Canada. [5]IRCCS Istituto Giannina Gaslini, Genoa, Italy. [6]Department of Maternal Infantile and Urological Sciences, Pediatric Onco-Hematology Unit, Policlinico Umberto I, Sapienza, University of Rome, Rome, Italy. [7]Copenhagen University Hospital – Rigshospitalet, Copenhagen, Denmark. [8]Department of Pediatric Hematology/Oncology/Hemostaseology, Center for Pediatric and Adolescent Medicine, University Medical Center of the Johannes Gutenberg University Mainz, Mainz, Germany. [9]Riley Hospital for Children, Indianapolis, IN, USA. [10]Hospital Infantil Universitario Niño Jesús, Madrid, Spain. [11]Hospital Universitario y Politecnico La Fe, Valencia, Spain. [12]Y-mAbs A/S, Hørsholm, Denmark. [13]Memorial Sloan Kettering Cancer Center, New York, NY, USA. ✉e-mail: jaume.mora@sjd.es

## Table 1 | Baseline characteristics in Trial 201

| Characteristic | Efficacy population (N = 52) | Safety population (N = 74) |
|---|---|---|
| **Patient characteristic** | | |
| Median age at enrollment, years (range) | 6 (2–18) | 6 (2–30) |
| **Sex, n (%)** | | |
| Male | 31 (60) | 45 (61) |
| Female | 21 (40) | 29 (39) |
| **Race, n (%)** | | |
| White | 21 (40) | 35 (47) |
| Asian | 29 (56) | 36 (49) |
| Black | 2 (4) | 2 (3) |
| Other | 0 (0) | 1 (1) |
| Median performance score, % (range)[a] | 100 (60–100) | 100 (60–100) |
| **Disease characteristic, n (%)** | | |
| *MYCN status* | | |
| Amplification | 7 (14) | 8 (11) |
| Gain | 1 (2) | 2 (3) |
| Neither gain nor amplification | 37 (71) | 57 (77) |
| Unknown | 7 (14) | 7 (10) |
| *INSS stage at diagnosis* | | |
| Stage 3 | 4 (8) | 6 (8) |
| Stage 4 | 46 (89) | 66 (89) |
| Unknown | 2 (4) | 2 (3) |
| *Histology (per INPC)* | | |
| Favorable | 2 (4) | 5 (7) |
| Unfavorable | 32 (62) | 47 (64) |
| Unknown | 18 (35) | 22 (30) |
| *Neuroblastoma location (independent review)* | | |
| Bone only | 29 (56) | 33 (45) |
| BM only | 2 (4) | 3 (4) |
| Bone and BM | 21 (40) | 23 (31) |
| *Baseline CS (independent review)[b]* | | |
| Median (range) | 3 (0–21) | 2 (0–21) |
| CS ≤ 2, n (%) | 22 (42) | 40 (55) |
| CS ≥ 3, n (%) | 30 (58) | 33 (45) |
| *Disease status* | | |
| Primary refractory | 26 (50) | 37 (50) |
| Relapsed | 26 (50) | 37 (50) |
| *Number of prior relapses[c]* | | |
| 1 relapse | 26 (50) | 37 (50) |
| Median time to 1st relapse, months (range)[d] | 20 (0–57) | 22 (0–57) |
| 2 relapses | 4 (8) | 8 (11) |
| Median time to 2nd relapse, months (range)[e] | 16 (9–21) | 21 (3–53) |
| Median time from last relapse to trial entry (months) | 6 (0.8[f]–25) | 6 (0.8–25) |
| **Prior treatment, n (%)** | | |
| *Treatment type* | | |
| Chemotherapy | 52 (100) | 73 (99) |
| Surgery | 46 (89) | 66 (89) |
| Radiotherapy | 21 (40) | 30 (41) |
| ASCT | 14 (27) | 22 (30) |
| Anti-GD2 therapy | 13 (25) | 20 (27) |
| MIBG | 4 (8) | 6 (8) |

Performance score was measured using the Lansky Play-Performance Scale (for patients age <16 years) or the Karnofsky Performance Status Scale (for patients age ≥16 years).
[a]Of the 66 patients with performance scores ≥90, 28 (44%) had a CS of ≥3.
[b]One patient in the safety analysis set had a missing baseline CS and was excluded from the subgroup analysis by CS.
[c]Relapsed patients had failed the salvage therapy administered for the relapse or PD prior to enrollment.
[d]Months from initial diagnosis to first relapse.
[e]Months from first relapse to second relapse.
[f]One patient was enrolled in violation with protocol requirement of 2-month interval between documented relapse and trial enrollment.
*ASCT* autologous stem cell transplant, *BM* bone marrow, *CS* Curie score, *INPC* International Neuroblastoma Pathology Committee, *INSS* International Neuroblastoma Staging System, *MIBG* meta-iodobenzylguanidine, SAF.

within and between patients over the course of treatment[12–15]. Despite advances in treatment for newly diagnosed patients with HR neuroblastoma[16–18], the prognosis remains poor, with an estimated 5-year overall survival (OS) rate of 50%[19]. Moreover, as many as 20% of patients are refractory to induction therapy, and >50% of patients experience relapse[16–23]. Although bone/BM acts as a frequent reservoir for chemoresistant neuroblastoma cells thought to drive refractory disease and relapse[8,14,24], previous studies have not evaluated treatment options for these compartments in isolation.

Naxitamab (previously called hu3F8) is a high-affinity, humanized monoclonal antibody (mAb) against GD2, a disialoganglioside implicated in the malignant transformation of neuroblastoma cells[25–29]. Naxitamab induces immune-mediated antitumor cytotoxicity, primarily via antibody-dependent cellular toxicity and complement-dependent cytotoxicity[27,30–32]. Granulocyte-macrophage colony-stimulating factor (GM-CSF) is co-administered with naxitamab to enhance its cytotoxic activity[33,34].

Naxitamab is approved in the United States (US), Israel, China, Brazil, and Mexico, among other countries, in combination with GM-CSF, for the treatment of pediatric (1 year of age and older) and adult patients with relapsed or refractory (R/R) HR neuroblastoma in the bone/BM who have demonstrated a partial response (PR), minor response (MR), or stable disease (SD) to prior therapy[35]. Preliminary efficacy results of naxitamab+GM-CSF from two independently conducted phase 2 clinical trials (Trial 12-230 [NCT01757626] and Trial 201 [NCT03363373]) in patients with R/R HR neuroblastoma and residual disease in bone/BM supported the accelerated US Food and Drug Administration approval for this vulnerable population[35].

In this prespecified interim analysis of Trial 201, we show that naxitamab demonstrates clinically meaningful efficacy with manageable safety in a well-defined population. Trial 201 was designed to investigate the efficacy and safety of naxitamab monotherapy (with GM-CSF) in patients with R/R HR-NB and residual disease in bone and/or BM only and excludes those with soft tissue lesions and actively relapsing/progressing disease who require more intensive therapies[36–38]. This study therefore comprised a narrowly defined subpopulation with residual disease in the bone/BM, reflecting the importance of targeting chemoresistant neuroblastoma cells and, for those with primary refractory disease, of achieving complete response (CR) prior to consolidation[10,39–41].

## Results

### Patient, disease, and treatment characteristics

Patient demographics and disease and treatment characteristics at enrollment are summarized in Table 1. Patient disposition is shown in Fig. 1, and per-site patient enrollment is shown in Supplementary Table 1. At that time, 74 patients had received the allocated treatment, with all 74 patients subsequently being evaluated for safety and 52 being evaluated for efficacy.

### Efficacy

In Trial 201, there were 26 responders and 26 non-responders, resulting in an overall response rate (ORR) of 50% (95% CI: 36–64%; $P < 0.0001$). CR and PR were observed in 38% (20/52) and 12% (6/52) of patients, respectively. Investigator-assessed ORR was 54% (95% CI: 40–68%; 28/52 patients). Median time to onset of response was 2 treatment cycles (range 2–8; ~7 weeks). Notably, 40% of responders who had PR ($n = 10$) as their initial response achieved CR ($n = 4$) with continued naxitamab treatment. Figure 2a, Tables 2–4, and Supplementary Table 2 show duration of response; response rates by disease status, response compartment, baseline disease location, baseline Curie score (CS), and prior anti-GD2 therapy; and response by *MYCN* status and sex, respectively. Patients with evaluable bone disease had a 58% (29/50 patients) bone-compartment response, with CR in 40% and

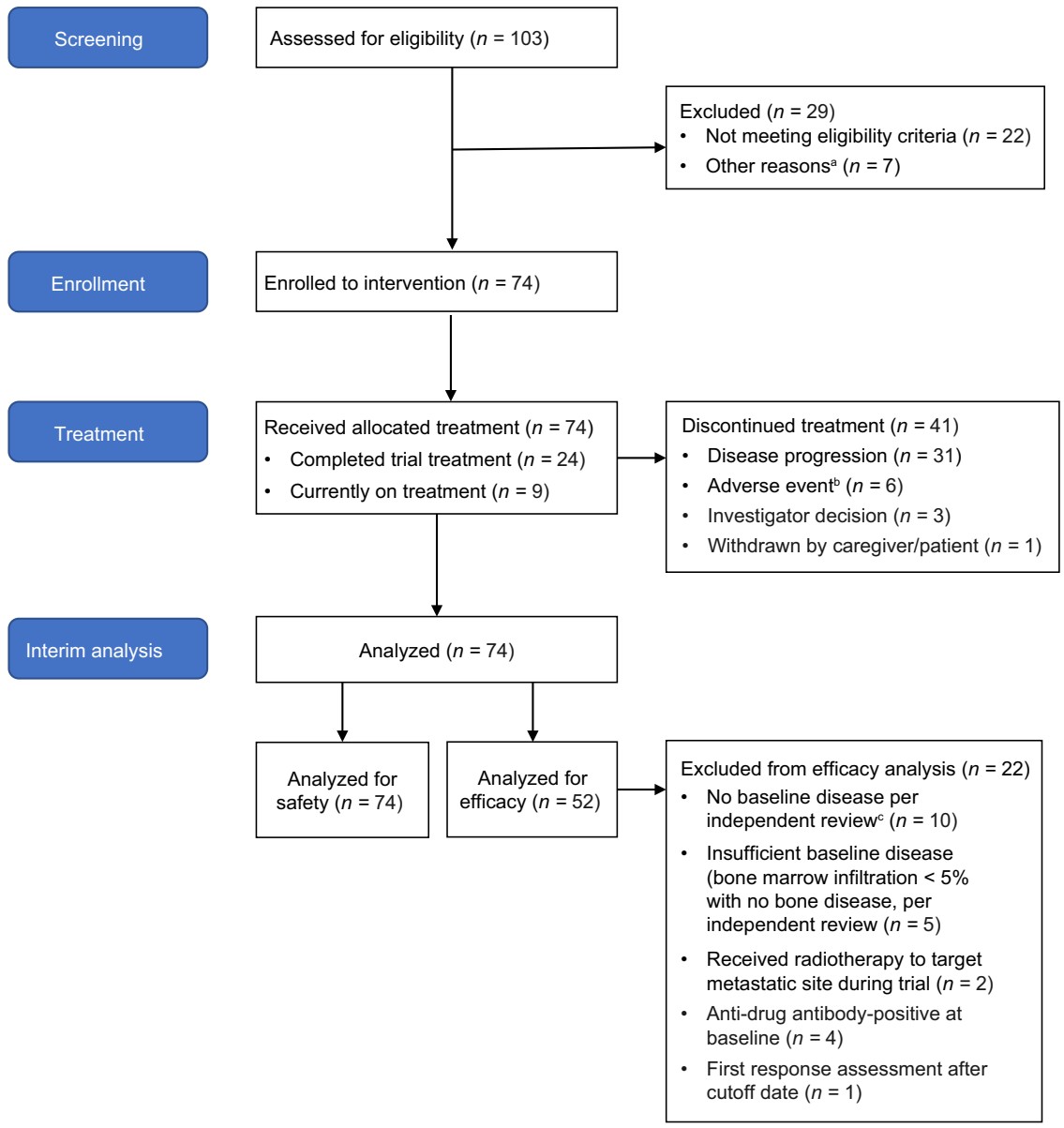

**Fig. 1 | Consort flow diagram for Trial 201.** [a]Other reasons include the following: did not meet required disease status or time from start of induction therapy to trial enrollment (*n* = 3); received immunosuppressive agents (*n* = 1); prior naxitamab treatment (*n* = 1); withdrawal of consent (*n* = 1); considered by investigator to be better suited for other treatment (*n* = 1). [b]Excluded due to the following: anaphylaxis (*n* = 2, SAEs, Grade 4); PRES (*n* = 1, SAE, Grade 3); respiratory depression (*n* = 1, SAE, Grade 4); hypotension (*n* = 1, non-serious, Grade 2, occurring in the same patient with respiratory depression leading to treatment discontinuation); urticaria (*n* = 1, SAE, Grade 2); myocarditis (*n* = 1, SAE, Grade 3, occurring in a patient with a history of hypertrophic cardiomyopathy). Some patients were excluded for more than one reason. [c]Excluded due to the following: reviewer agreement to CS = 0 and no BM involvement (*n* = 2); reviewer agreement to CS = 1 in retroperitoneal soft tissue (not spine) (*n* = 1); reviewer disagreement to CS = 0, with third reviewer confirming CS = 0 (*n* = 7); no BM samples available for independent review (*n* = 3). Some patients were excluded for more than one reason. BM bone marrow, CS Curie score, PRES posterior reversible encephalopathy syndrome, SAE serious adverse event, TEAE treatment-emergent adverse event.

PR in 18% of patients. BM compartment response was 74% (17/23 patients) with CR in 74% of patients. Of the 13 patients with prior exposure to anti-GD2 mAbs, all but one (92%) had relapsed disease, and 4 (31%) responded to naxitamab, 2 (50%) of whom received anti-GD2 therapy as part of their most recent treatment regimen prior to Trial 201 enrollment. Among the 39 patients with no prior anti-GD2 therapy, 14 (36%) had relapsed disease.

The 52 patients completed a median of 7 naxitamab treatment cycles (range 1–20). The median duration of efficacy follow-up from start of treatment was 6 months (range 1–18), and median duration of long-term follow-up was 21 months (range 1–44). Median duration of response (DoR) was not estimable (NE; 95% CI: 25 weeks to NE), as 77% (20/26) of patients with CR or PR had ongoing responses at the last

available assessment (independent review; *n* = 8) or were censored due to initiation of new anti-cancer treatment (*n* = 12, with 6 due to consolidation of the CR with an investigational GD2/GD3 vaccine; Fig. 2a)[42]. In patients with baseline CS ≤ 2, ORR was 50% (95% CI: 28–72%), with 45% achieving CR. In patients with baseline CS ≥ 3, ORR was 50% (95% CI: 31–69%), with 33% achieving CR. For patients with evaluable disease in bone, median CS was 3 (range 1–20) at baseline. The median absolute CS change (reduction) from baseline to maximum change was −2 (*n* = 48; range −18 to 14) with a median percentage CS change (reduction) of 69%. Individual maximum changes in CS are shown in Fig. 2b. Of the 50 patients with baseline evaluable bone disease, 2 did not have end of treatment (EOT) CSs and were not included in the CS analysis.

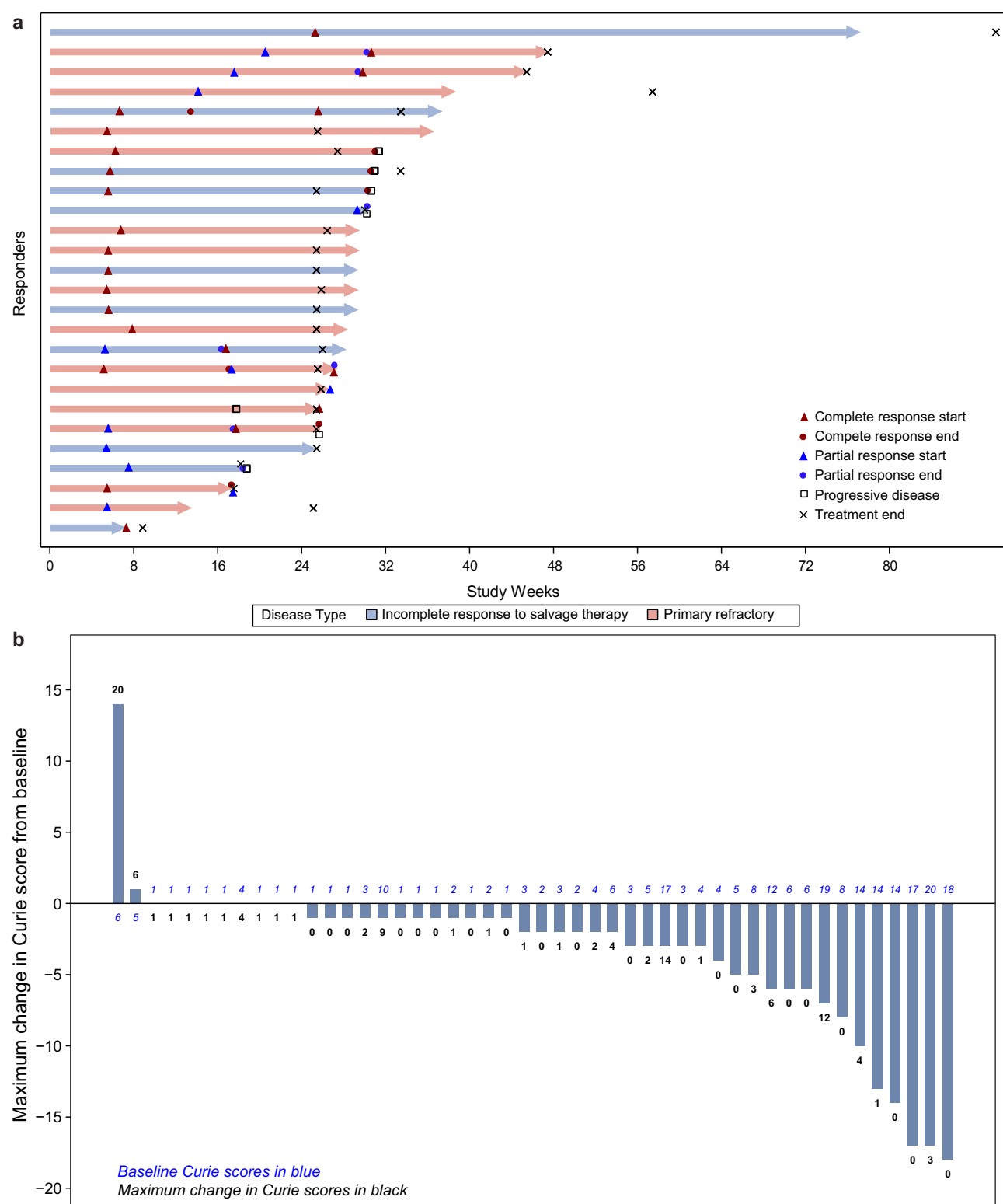

**Fig. 2 | Efficacy assessment of naxitamab. a** Duration of response as per independent review for each of the responders in the efficacy population (*n* = 26). **b** Maximum change in CS from baseline in patients with baseline disease in the bone compartment (*n* = 48; 2 of the 50 patients with evaluable bone disease at baseline did not have postbaseline CS assessment). Source data are provided as a Source Data file. CS Curie score.

The 1-year PFS was 35% (95% CI: 16–54%); median PFS was 30 weeks (95% CI: 18 to NE; Fig. 3a). The 1-year OS was 93% (95% CI: 80–98%), and median OS was NE, as 46/52 (88%) patients were alive at the time of data cutoff (Fig. 3b).

**Safety**

The 74 patients in the safety population completed a median of 7 naxitamab treatment cycles (range 0–20). No toxicity to major organs (including the liver, kidneys, heart, and lungs) was observed, and the

**Table 2 | Best responses overall and by disease status in 52 patients treated with naxitamab plus GM-CSF**

| Patient population | Endpoint | Patients | |
|---|---|---|---|
| | | n (%) | 95% CI, % |
| Overall (N = 52) | CR | 20 (39) | 25–53 |
| | PR | 6 (12) | 4–23 |
| | MR | 5 (10) | 3–21 |
| | SD | 10 (19) | 10–33 |
| | PD | 8 (15) | 7–28 |
| | NE | 3 (6) | – |
| ORR | CR + PR | 26 (50) | 36–64 |
| Disease status | | | |
| Refractory neuro-blastoma (n = 26) | CR | 12 (46) | 27–67 |
| | PR | 3 (12) | 2–30 |
| | MR | 3 (12) | 2–30 |
| | SD | 3 (12) | 2–30 |
| | PD | 4 (15) | 4–35 |
| | NE | 1 (4) | – |
| ORR | CR + PR | 15 (58) | 37–77 |
| Relapsed neuro-blastoma (n = 26) | CR | 8 (31) | 14–52 |
| | PR | 3 (12) | 2–30 |
| | MR | 2 (8) | 1–25 |
| | SD | 7 (27) | 11–48 |
| | PD | 4 (15) | 4–35 |
| | NE | 2 (8) | – |
| ORR | CR + PR | 11 (42) | 23–63 |

Best response according to independent review, based on INRC (efficacy population).
*CR* complete response, *GM-CSF* granulocyte-macrophage colony-stimulating factor, *INRC* International Neuroblastoma Response Criteria, *MD* minimal disease, *MR* minor response, *NE* not evaluable, *ORR* overall response rate, *PD* progressive disease, *PR* partial response, *SD* stable disease.

**Table 3 | Best responses by evaluable disease in bone/BM compartment and disease location at enrollment in 52 patients treated with naxitamab plus GM-CSF**

| Patient population | Endpoint | Patients | |
|---|---|---|---|
| | | n (%) | 95% CI, % |
| **Response by evaluable disease in bone/BM compartment[a]** | | | |
| Evaluable disease in BM (n = 23) | CR | 17 (74) | 52–90 |
| | MD | 1 (4) | 0–22 |
| | SD | 2 (9) | 1–28 |
| | PD | 1 (4) | 0–22 |
| | NE | 2 (9) | – |
| Response rate | CR + PR | 17 (74) | 52–90 |
| Evaluable disease in bone (n = 50) | CR | 20 (40) | 26–55 |
| | PR | 9 (18) | 9–31 |
| | SD | 13 (26) | 15–40 |
| | PD | 6 (12) | 5–24 |
| | NE | 2 (4) | – |
| Response rate | CR + PR | 29 (58) | 43–72 |
| **Response by disease location at enrollment** | | | |
| BM disease only (n = 2) | CR | 1 (50) | 1–99 |
| | PD | 1 (50) | 1–99 |
| ORR | CR + PR | 1 (50) | 1–99 |
| Bone disease only (n = 29) | CR | 10 (35) | 18–54 |
| | PR | 5 (17) | 6–36 |
| | SD | 7 (24) | 10–44 |
| | PD | 7 (24) | 10–44 |
| | NE | 0 | – |
| ORR | CR + PR | 15 (52) | 33–71 |
| Both bone and BM disease (n = 21) | CR | 9 (43) | 22–66 |
| | PR | 1 (5) | 0–24 |
| | MR | 5 (24) | 8–47 |
| | SD | 3 (14) | 3–36 |
| | NE | 3 (14) | – |
| ORR | CR + PR | 10 (48) | 26–70 |

Best response according to independent review, based on INRC (efficacy population).
[a]Subjects may provide data to both categories.
*BM* bone marrow, *CR* complete response, *GM-CSF* granulocyte-macrophage colony-stimulating factor, *INRC* International Neuroblastoma Response Criteria, *MD* minimal disease, *MR* minor response, *NE* not evaluable, *ORR* overall response rate, *PD* progressive disease, *PR* partial response, *SD* stable disease.

trial treatment did not affect performance scores. All patients experienced ≥1 treatment-emergent adverse event (AE), 81% of which were treatment related. Most related AEs (90%) were infusion-related reactions occurring on an infusion day after the start of a naxitamab infusion. Table 5 summarizes the most frequently reported treatment-related AEs of any grade, and the corresponding treatment-related AEs of Grade 3-4. Of the reported treatment-related AEs of Grade 3-4, 54% (419/769 events) lasted <1 hour, with 83% lasting ≤5 hours. AEs of hypotension and bronchospasm were reported frequently, with most observed during Cycle 1. Of the 201 reported Grade 3-4 hypotension events, the majority (70%) resolved in ≤1 hour. Grade 3-4 infusion-related hypotension (including 3 Grade 4 AEs in 3 patients) was experienced by 47% of patients in Cycle 1 vs 27% in Cycle 7, and Grade 3 bronchospasm (no Grade 4 AEs) by 6.8% of patients in Cycle 1 vs 2.4% in Cycle 7. At data cutoff, 98% (815/834 events) of all Grade 3-4 AEs had resolved.

Grade 3-4 AEs of pain were reported in 54% of patients, with decreasing frequency across treatment cycles: Grade 3 pain was experienced by 53% of patients in Cycle 1 and 37% in Cycle 7. During infusions in Cycles 1 and 2, the median Face, Legs, Activity, Cry, Consolability (FLACC) score for worst pain was 8 (range 0–10), and median Wong–Baker FACES scores were 8 and 6, respectively (Supplementary Table 3). For both scales, by 15 minutes after infusion, median scores returned to 0 and remained at 0 until discharge. Of the 412 reported Grade 3 pain events, 52% resolved in ≤1 hour (Supplementary Table 4).

Thirty-three patients experienced 50 treatment-emergent serious adverse events (TESAEs), of which 31 (62%) were considered treatment related. TESAEs reported in ≥5% of patients included hypotension (7%), device-related infection (7%), pyrexia (5%), and rash (5%). Treatment-related serious adverse events (SAEs) reported in ≥5% of patients

included hypotension (7%) and rash (5%). Six patients discontinued naxitamab treatment due to treatment-related AEs (Supplementary Table 5). There were no AEs of transverse myelitis or fatal treatment-related AEs.

The trial protocol provides guidance for managing AEs during naxitamab infusion, including recommendations for supportive therapies readily available at bedside (Supplementary Fig. 1) and for infusion rate reductions or pauses[43]. Consistent with local pain mitigation strategies, 13 patients received ketamine as premedication. Glucocorticoids were not initially required as premedication for Infusion 1, Cycle 1, but were later required by a protocol amendment. As a result, 20 patients (27%) did not receive glucocorticoids. Importantly, among patients who received glucocorticoid premedication during Infusion 1, Cycle 1, versus those who did not, there was a lower frequency of Grade ≥3 bronchospasm (3.7% vs 15.0%), urticaria (5.6% vs 20.0%), and anaphylactic reaction (0% vs 15.0%). Of the 4 SAEs of Grade 3-4 anaphylactic reaction in patients who had not received glucocorticoid premedication, 3 events occurred in Cycle 1, and 1 event occurred in Cycle 2.

During the trial, 92% (1144/1237) of naxitamab infusions were administered in the outpatient setting, without the need for an

**Table 4 | Best responses by CS at enrollment and prior anti-GD2 mAb treatment in 52 patients treated with naxitamab plus GM-CSF**

| Patient population | Endpoint | Patients | |
|---|---|---|---|
| | | n (%) | 95% CI, % |
| **Response by CS at enrollment** | | | |
| **CS ≤ 2 (n = 22)** | CR | 10 (45) | 24–68 |
| | PR | 1 (5) | 0–23 |
| | MR | 2 (9) | 1–29 |
| | SD | 7 (32) | 14–55 |
| | PD | 2 (9) | 1–29 |
| **ORR** | CR + PR | 11 (50) | 28–72 |
| **CS ≥ 3 (n = 30)** | CR | 10 (33) | 17–53 |
| | PR | 5 (17) | 6–35 |
| | MR | 3 (10) | 2–27 |
| | SD | 3 (10) | 2–27 |
| | PD | 6 (20) | 8–39 |
| | NE | 3 (10) | – |
| **ORR** | CR + PR | 15 (50) | 31–69 |
| **Response by prior anti-GD2 mAb** | | | |
| **No prior anti-GD2 mAb (n = 39)** | CR | 17 (44) | 28–60 |
| | PR | 5 (13) | 4–27 |
| | MR | 4 (10) | 3–24 |
| | SD | 6 (15) | 6–31 |
| | PD | 4 (10) | 3–24 |
| | NE | 3 (8) | – |
| **ORR** | CR + PR | 22 (56) | 40–72 |
| **Prior anti-GD2 mAb (n = 13)** | CR | 3 (23) | 5–54 |
| | PR | 1 (8) | 0–36 |
| | MR | 1 (8) | 0–36 |
| | SD | 4 (31) | 9–61 |
| | PD | 4 (31) | 9–61 |
| **ORR** | CR + PR | 4 (31) | 9–61 |

Best response according to independent review, based on INRC (efficacy population).
CR complete response, CS Curie score, GM-CSF granulocyte-macrophage colony-stimulating factor, INRC International Neuroblastoma Response Criteria, mAb monoclonal antibody, MD minimal disease, MR minor response, NE not evaluable, ORR overall response rate, PD progressive disease, PR partial response, SD stable disease.

overnight stay, and 7.5% (93/1237) took place in an inpatient setting. The planned dose of naxitamab (3 mg/kg/day) was delivered in 99% (1224/1237) of infusions. The median infusion time was 37 minutes (range 7–175 minutes).

## Discussion

HR neuroblastoma is a singularly complex and heterogeneous pediatric malignancy with current and emerging research investigating the role of targeted therapy for select populations[13,44]. In Trial 201, we investigated the efficacy and safety of treatment with naxitamab plus GM-CSF in patients with R/R HR neuroblastoma with residual disease limited to the bone and/or BM, a well-defined segment of the overall HR neuroblastoma population in urgent need of efficacious treatment options with manageable safety profiles.

Results from this prespecified interim analysis show statistically significant and clinically meaningful outcomes, with an ORR of 50% (95% CI: 36–64%) and a CR rate of 38%. With the lower limit of the 95% exact CI for ORR in the efficacy population exceeding 20%, the trial met its primary endpoint as specified in this interim analysis. Efficacy was also concluded from the investigator-assessed ORR (54%; 95% CI: 40–68%), which was notably similar to the external independent assessment.

The study was not powered for subgroup analyses, limiting any firm conclusions. Several preliminary findings warrant discussion and additional research. First, ORR was 58% in patients with refractory disease and 42% in patients with relapsed disease, which are findings consistent with previous studies demonstrating that patients with refractory neuroblastoma have responded to various treatments at higher rates than those with relapsed disease[16,17]. Further, there were no apparent differences in outcomes among patients with *MYCN* amplification, although the subgroup was underrepresented in this trial population (14%) compared with a higher estimated prevalence of ≤25% in all neuroblastoma cases[5,45] and ≤40% in patients with HR disease[46]. *MYCN* amplification is an established oncogenic driver associated with poor prognosis, underscoring the need for further investigation and caution in interpreting the generalizability of these findings.

In this study, 31% of patients demonstrated a response after having previously received another anti-GD2 mAb (Table 4). Little is known about the differential response to anti-GD2 mAbs in patients with HR neuroblastoma, an issue of growing importance with dinutuximab and dinutuximab beta, which are approved for use in HR neuroblastoma in the US and Europe, respectively[36,47–51]. While additional research is needed, the observed response to naxitamab in patients previously treated with an anti-GD2 mAb may reflect differences in their respective pharmacokinetic/pharmacodynamic properties, as well as patient and disease heterogeneity.

Median DoR was NE in the study population. Of the 20 patients censored, 8 patients maintained their response at the last independent review, and 12 were censored for initiating new anti-cancer treatment, including 6 patients who received an investigational GD2/GD3 vaccine for consolidation of the CR. Preliminary results for the long-term secondary endpoint are promising, with a 1-year OS rate of 93%. Long-term follow-up will establish the 3-year OS rate.

Trial 201 investigated the efficacy of naxitamab in patients with disease in the well-vascularized compartments of bone and BM, a common metastatic niche for chemoresistant neuroblastoma cells closely linked to poor outcomes[8,14,24,52–54]. The high response rates to naxitamab in bone (58%) and BM (74%), therefore, have important implications for patient outcomes, underscored by an ORR of 50% across the R/R setting. Notably, clinically meaningful responses were observed irrespective of baseline CS (≤ 2 vs ≥3), with marked CS reductions of up to 18. These results therefore demonstrate the efficacy of single-agent naxitamab ( + GM-CSF) in patients with metastatic bone/BM involvement. It is important to note, however, that neuroblastoma tumors employ elaborate immune evasion mechanisms, creating a tumor microenvironment that may limit the efficacy of single-agent anti-GD2 therapy[12,19,55,56]. In a seminal study of patients with R/R HR neuroblastoma, dinutuximab in combination with irinotecan and temozolomide (DIT) showed limited efficacy in patients with soft tissue lesions (objective response 21.6%) vs those with evaluable disease in bone/BM (objective response 87.5%), supporting chemotherapy alone or in combination with anti GD2 mAbs as the standard of care for the treatment of patients with metastatic soft tissue lesions[36]. To assess the impact of naxitamab monotherapy ( + GM-CSF) without the potentially confounding effects of concurrent chemotherapy, Trial 201 therefore excluded patients with soft tissue disease at enrollment. Similarly, the study population did not include patients with active progressive disease (PD) who require intensive, multimodal treatment regimens[38].

Safety outcomes for naxitamab from this interim analysis are consistent with those previously reported[43]. No AEs of transverse myelitis or fatal treatment-related AEs were reported. The most commonly reported related Grade 3-4 AEs with the potential to cause severe clinical complications were hypotension and bronchospasm, although most (70%) of the Grade 3-4 hypotension events resolved quickly (≤1 hour). The frequency of Grade 3 pain and bronchospasm

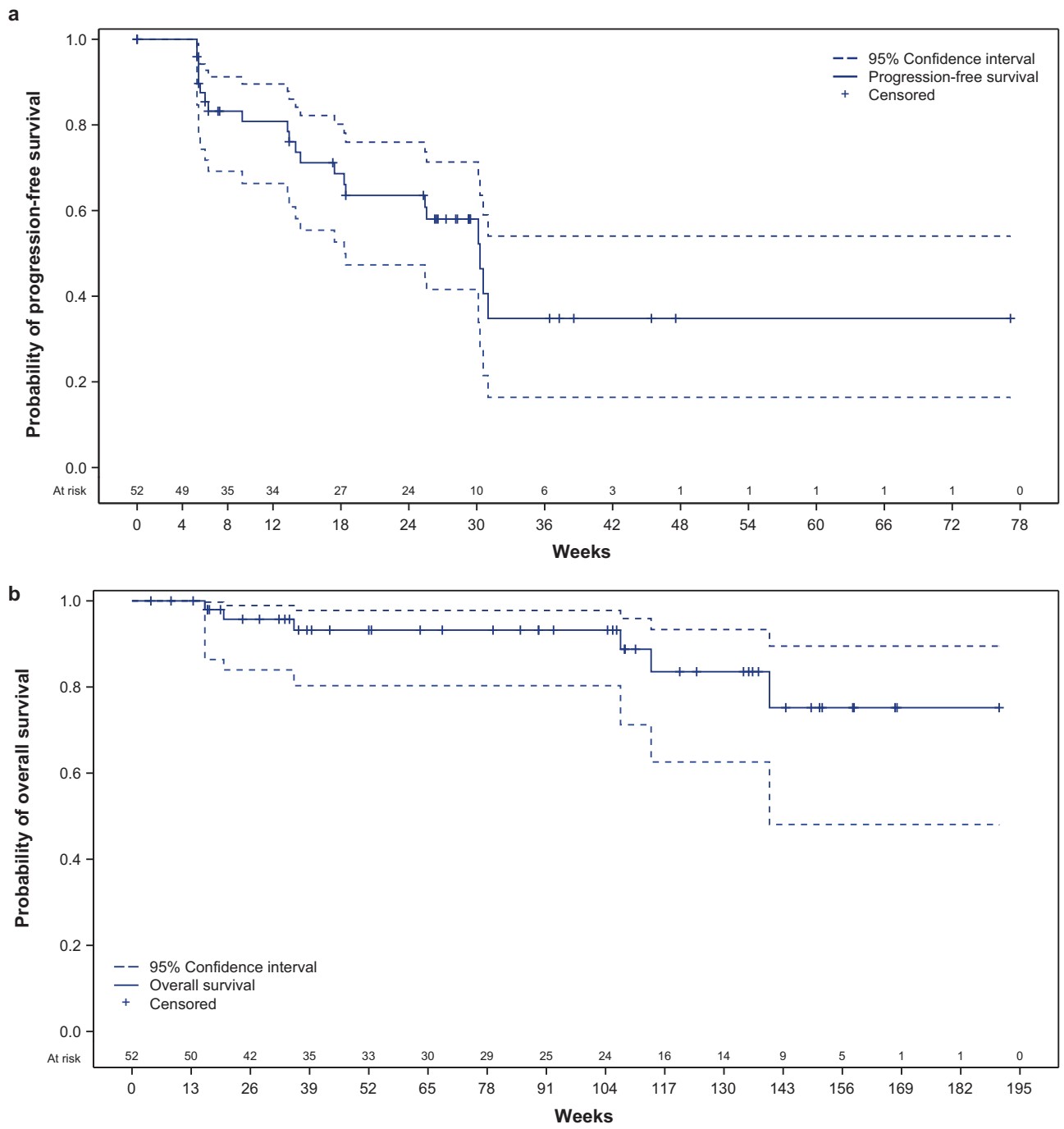

**Fig. 3 | Kaplan–Meier estimates for (a) PFS and (b) OS for patients treated with naxitamab plus GM-CSF in Trial 201 at data cutoff (December 31, 2021). Efficacy population, *n* = 52. a** PFS is defined as the time from the first infusion of naxitamab until PD or death, whichever comes first, and censoring occurs at the earliest of the date of last disease evaluation before initiation of new anti-neuroblastoma treatment or last assessment during long-term follow-up. **b** OS is defined as the time from the first infusion of naxitamab until death, and censoring occurs at the last date where the subjects are known to be alive during long-term follow-up. Source data are provided as a Source Data file. GM-CSF granulocyte-macrophage colony-stimulating factor, OS overall survival, PD progressive disease, PFS progression-free survival.

and Grade 3-4 infusion-related hypotension demonstrated reductions across treatment cycles, a change in trajectory of AEs that may inform mitigation strategies. Grade 3 pain was reported frequently, a known class effect of anti-GD2 mAb therapy due to GD2 expression on peripheral nerve fibers[48,57,58]. Although pain during infusions was intense, it was short lived, dissipating within 15 minutes of infusion completion, and it did not lead to treatment discontinuation for any patient. The AE most commonly leading to treatment discontinuation was anaphylactic reaction, which occurred in 2 of the 6 patients who

discontinued trial treatment due to treatment-related AEs, both of whom had not received glucocorticoid premedication. Treatment discontinuation due to treatment-related hypotension occurred in 1 patient.

Although naxitamab is associated with a high frequency of severe infusion-related reactions, its safety profile is manageable with mitigations in place. As shown in this study, the majority of naxitamab infusions (92%) were administered in the outpatient setting; median infusion time was 37 minutes; all infusions were completed in <3 hours;

**Table 5 | Related AEs reported in ≥ 10% of patients in Trial 201 and associated related Grade 3 and Grade 4 AEs (safety population)**

| SOC | PT | Patients (N = 74) | | |
|---|---|---|---|---|
| | | Related[a] AEs Any grade, n (%) | Related AEs Grade 3, n (%) | Related AEs Grade 4, n (%) |
| General disorders and administration site conditions | Pain | 69 (93) | 40 (54) | N/A[b] |
| | Pyrexia | 48 (65) | 1 (1) | 0 |
| | Fatigue | 14 (19) | 1 (1) | N/A[b] |
| | Face edema | 13 (18) | 1 (1) | N/A[b] |
| | Edema | 11 (15) | 0 | 0 |
| | Chest pain | 9 (12) | 1 (1) | N/A[b] |
| Vascular disorders | Hypotension | 72 (97) | 43 (58) | 3 (4) |
| | Hypertension | 21 (28) | 4 (5) | 0 |
| Skin and sub-cutaneous tissue disorders | Urticaria | 54 (73) | 14 (19) | N/A[b] |
| | Pruritus | 32 (43) | 1 (1) | 0 |
| | Rash | 16 (22) | 3 (4) | 0 |
| Gastrointestinal disorders | Abdominal pain | 39 (53) | 12 (16) | N/A[b] |
| | Nausea | 28 (38) | 0 | 0 |
| | Vomiting | 27 (37) | 0 | 0 |
| | Diarrhea | 24 (32) | 0 | 0 |
| Respiratory, thoracic, and mediastinal disorders | Bronchospasm | 43 (58) | 13 (18) | 0 |
| | Cough | 29 (39) | 0 | 0 |
| | Hypoxia | 11 (15) | 7 (9) | 0 |
| | Wheezing | 9 (12) | 0 | 0 |
| Cardiac disorders | Tachycardia | 37 (50) | 2 (3) | 0 |
| | Sinus tachycardia | 15 (20) | 0 | N/A[b] |
| Musculoskeletal and connective tissue disorders | Pain in extremity | 29 (39) | 1 (1) | N/A[b] |
| Investigations | Oxygen saturation decreased | 13 (18) | 0 | 0 |
| | Alanine amino-transferase increased | 8 (11) | 5 (7) | 0 |
| Nervous system disorders | Headache | 13 (18) | 4 (5) | N/A[b] |
| Metabolism and nutrition disorders | Decreased appetite | 12 (16) | 0 | 0 |
| Immune system disorders | Hypersensitivity | 10 (14) | 2 (3) | 0 |
| Eye disorders | Mydriasis | 10 (14) | 0 | 0 |

[a]Naxitamab- or naxitamab+GM-CSF-related AE.
[b]These events are of Grade 3 or lower as per CTCAE version 4.0.
*AE* adverse event, *CTCAE* Common Terminology Criteria for Adverse Events, *GM-CSF* granulocyte-macrophage colony-stimulating factor, *N/A* not applicable, *PT* preferred term, *SOC* system organ class.

and 99% of infusions provided the planned dose of 3 mg/kg/day. Administration of naxitamab, regardless of location, requires a coordinated and prepared multidisciplinary team. As recently reported by Trovillion et al.[59], a well-planned approach for close monitoring, timely recognition, and immediate intervention has been shown to enhance the treatment experience for both patients and providers. Implementing strategies for emergency response and respiratory support and managing adverse events with premedication are of particular importance[14,43,60,61].

The wide list of supportive bedside medications shown in Supplementary Fig. 1 allow for swift treatment of some of the most common acute side effects associated with naxitamab, including anaphylaxis, bronchospasm, hypotension, nausea/vomiting, and pain. Consensus guidelines on pain management strategies during naxitamab administration are available for consideration[57].

The findings from this interim analysis demonstrate clinically meaningful efficacy for naxitamab, alongside a safety profile deemed acceptable for these at-risk patients, providing additional evidence supporting anti-GD2 immunotherapy in targeted subpopulations with HR neuroblastoma[36,37,62–64].

Trial 201 is a global, single-arm, open-label clinical trial that included patients with R/R HR neuroblastoma in the bone and/or BM compartments only, where single-drug therapy was thought to be most effective[31,65,66]. Further research is required to explore the effects of naxitamab and other treatment modalities on patients not included in this relatively homogeneous cohort. Patients with soft tissue disease or active PD, in particular, were excluded from this trial and may benefit from combined treatment with chemotherapeutic agents and anti-GD2 mAbs, as previously reported[36,51,67].

Consistent with pre-specified criteria, a significant proportion of patients ($n = 22$) were excluded from the efficacy population based on a centralized, independent, and retrospective analysis of baseline data, although all were included in the safety analysis set and, therefore, contributed to our understanding of the naxitamab safety profile. Of the 22 excluded patients, 10 had no baseline evaluable disease, and 5 patients had insufficient baseline BM disease, which did not meet threshold requirements for gauging treatment response. Four patients with positive anti-drug antibody status were also excluded from the efficacy analysis to minimize the potential confounds associated with the complex and still poorly understood relationship between anti-drug antibodies and treatment response. Protocol criteria further excluded patients with Karnofsky/Lansky scores of <50%, and of the 66 patients with performance scores ≥90, 28 (44%) had a CS of ≥3. Although few if any studies have investigated performance scores in patients with HR neuroblastoma, it is plausible that the performance scores of patients with actively progressing disease would on average be lower, and their disease burdens correspondingly higher, compared with those included in Trial 201. Additional research is required to better characterize the relationship between performance scores and disease burden across the patient journey.

Taken together, the inclusion and exclusion criteria define a carefully selected subpopulation of patients with residual disease in the bone/BM, which inherently limits the generalizability of the results across a varied population of patients with HR neuroblastoma with distinct treatment needs, historical outcomes, and tumor and disease burdens.

This is a report of results from a prespecified interim analysis. The resulting potential loss of equipoise may affect ongoing patient enrollment and treatment and may have an impact on the final analysis. These risks are perhaps best measured against the need for sharing the data with the clinical community in a peer-reviewed forum. The final trial analysis will include a larger patient population and provide efficacy and safety data on long-term outcomes, including OS. This final analysis will use the same 2-sided 95% CI as the interim analysis, with no planned adjustment for multiplicity.

Although the International Neuroblastoma Risk Group (INRG) staging system is the current accepted standard for disease staging, this study was designed to align with the methodology from the Phase 1/2 trial conducted in 2012 (ISS12-230), which utilized the International Neuroblastoma Staging System (INSS)[58]. Most patients (89%) in both the safety and efficacy populations were INSS stage 4, which is equivalent to INRG stage M[68]. In addition, the requirement for HR disease at the time of diagnosis per inclusion criteria is classified per the INRG staging system[69].

Refractory disease was defined as an incomplete response (PR or worse) to induction therapy, which is not aligned with enrollment criteria in trials sponsored by the Children's Oncology Group (COG)

and the International Society of Paediatric Oncology European Neuroblastoma (SIOPEN) Research Network. Both cooperative groups continue to evaluate optimal end-of-induction (EOI) cut points for patients enrolled in their respective HR neuroblastoma trials[40,41,70]. Independent retrospective studies have suggested that an EOI CS of ≤2 or SIOPEN score of ≤3 is associated with better long-term outcomes, and a retrospective analysis by Streby et al. recently demonstrated an optimal EOI CS of 0[39,41,48]. The evolving literature thus highlights the need for additional research on EOI CSs, as part of a broader effort to define refractory disease and the prognostic value of an EOI CR. The interim results presented here provide additional insights into this important issue.

Naxitamab is a humanized IgG1 mAb engineered to bind with high affinity and specificity to the tumor-associated antigen GD2[28]. Naxitamab plus GM-CSF is an effective treatment for patients with R/R HR neuroblastoma with residual disease in the bone/BM after induction or after treatment for progressive or relapsed disease. Without the use of concurrent chemotherapy, naxitamab showed responses in half of patients and complete responses in 38%, addressing a significant unmet need in this targeted subpopulation. Considering the gravity of HR neuroblastoma, naxitamab demonstrated an acceptable safety profile, with AEs that can be effectively managed in outpatient settings within a dedicated multidisciplinary team supported by established processes and treatment algorithms[43].

## Methods

The trial was conducted in accordance with applicable regulatory requirements, the International Council for Harmonization (ICH) Good Clinical Practice (GCP), and the ethical principles of the Declaration of Helsinki. Individual sites obtained appropriate Institutional Review Board approval (Supplementary Table 5). Written informed consent was obtained from legal guardian(s) and/or patient in accordance with local regulations. Trial 201 patients have only been compensated for travel as agreed with participating sites and described in the Clinical Trial Agreement and site-specific informed consent form. This study was registered on ClinicalTrials.gov under the identifier NCT03363373 on December 5, 2017.

Trial 201 is a global, single-arm, non-randomized, open-label, phase 2 trial designed to evaluate the efficacy and safety of naxitamab administered with GM-CSF in patients with R/R HR neuroblastoma as defined by INRG criteria.[69] Electronic case report forms were used to capture trial results and data. Source data for all data presented in graphs within figures is included. A redacted study protocol, amendments, and deviations are available in the Supplementary Information.

### Patient population

The first patient in the pre-planned interim data set was enrolled on April 11, 2018, and the last patient was enrolled on December 15, 2021. Patient recruitment and data collection began on April 23, 2018. The data cutoff date for this interim analysis was December 31, 2021. Disease and patient heterogeneity required careful selection of the Trial 201 study population to determine the efficacy and safety of naxitamab when administered as a single-agent therapy (i.e., not combined with chemotherapy). Patients were classified as having refractory or relapsed neuroblastoma at enrollment. Patients classified as refractory were those with an incomplete response (PR, MR, or SD per 2017 International Neuroblastoma Response Criteria [INRC][71]) to initial (induction) therapy and residual disease in the bone/BM only. All patients were enrolled within 18 months of initiating chemotherapy and had received ≥4 cycles of standard induction chemotherapy regimens comparable to one of the following protocols: COG ANBL0532 (NCT00567567) or A3973 (NCT00004188); MSKCC N6[72] or N7 (NCT00002634)[73]; or rapid cisplatin, vincristine, carboplatin, etoposide, and cyclophosphamide (COJEC)[74].

Patients classified as relapsed were those with an incomplete response to treatment for actively progressing or relapsed disease with residual disease in the bone/BM only as per 2017 INRC. To distinguish refractory disease from PD or relapsed disease, a minimum of 2 months from documented PD or relapse was required, allowing time for patients to receive and respond to therapy prior to study enrollment.

Patients were required to have acceptable hematological status at screening, including hemoglobin ≥8 g/dL, white blood cell count ≥1000/μL, absolute neutrophil count of ≥500/μL, and platelet count ≥25,000/μL. Patients with actively progressing disease at trial entry according to the 2017 INRC[71] were excluded. Additional exclusion criteria included prior treatment with naxitamab, any systemic anticancer therapy within 3 weeks of the first dose of GM-CSF, evaluable neuroblastoma outside bone and BM, and existing major organ dysfunction or active life-threatening infection.

Patients were enrolled at the investigator's discretion, and an independent central committee of board-certified radiologists and pathologists subsequently reviewed each patient's baseline data set as per INRC 2017. Patients who did not fulfill protocol-specified criteria for evaluable disease were retrospectively excluded from the efficacy population and included in the safety population. Supplementary Table 6 presents full inclusion and exclusion criteria. Sex was determined based on self-report or parental report. The distribution of male and female patients is reported in Table 1.

### Treatment

Each cycle of immunotherapy comprised subcutaneous GM-CSF at 250 μg/m²/day on Days −4 to 0, followed by 500 μg/m²/day on Days 1 to 5 (Supplementary Fig. 2a). Naxitamab 3 mg/kg/day was infused intravenously on Days 1, 3, and 5, with GM-CSF given at least an hour prior (Supplementary Table 7). Specifically, naxitamab was infused over at least 60 minutes during the first infusion and over at least 30 minutes during all subsequent infusions (infusion duration could be increased at the investigator's discretion to manage adverse reactions). GM-CSF was withheld if the white blood cell count was >50×10⁹/L or the absolute neutrophil count was >20×10⁹/L.

Treatment cycles were repeated every 4 weeks until a patient had a CR or PR, which was followed by 5 additional cycles. Subsequent cycles could be repeated every 8 weeks for up to 101 weeks after the first infusion at the discretion of the treating physician (Supplementary Fig. 1b). Long-term follow-up took place quarterly following the EOT visit and continued for up to 3 years. The end of the trial was defined as the last subject's last visit in the long-term follow-up period.

Prior to naxitamab infusions, premedication with normal saline fluid bolus, glucocorticoids (only mandated before the first naxitamab dose Cycle 1), antihistamine, gabapentin, and opioids (with a preference for oral administration to reduce the risk of hypotension during naxitamab infusion) was required to mitigate pain and infusion-related AEs, as described in detail by Mora et al.[43]. For breakthrough pain, intravenous opioids and/or ketamine were given at the discretion of the treating physician. Site initiation meetings were conducted to ensure the sharing of best practices and the safe and effective administration of infusions and use of pre-medications, supportive therapies, and AE-management algorithms according to each site's established processes and procedures (Supplementary Table 8, Supplementary Fig. 1).

### Efficacy assessment

The primary endpoint was overall response per 2017 INRC (Supplementary Tables 8 and 9) during the naxitamab treatment period, with disease response in bone and BM assessed between Cycles 2 and 3, and at pre-specified time points thereafter. Responders demonstrated an overall response of CR or PR. Patients with INRC response classification

of MR, SD, or PD, or those who were not evaluable after baseline were considered non-responders[71].

Independent central reviewers who were blinded to the clinical information evaluated responses per 2017 INRC. Imaging was based on computed tomography (CT) or magnetic resonance imaging (MRI) and [123]I-meta-iodobenzylguanidine (MIBG) scans; alternatively, [123]I-MIBG-single photon emission CT (MIBG-SPECT) could be used. [18]F-Fluorodeoxyglucose-positron emission tomography (FDG-PET) could be used for MIBG-non-avid neuroblastoma. Bilateral BM aspirates and biopsies were assessed by histopathology and immunohistochemistry using antibodies targeting synaptophysin[71]. The distinction between bone, BM, and bone/BM disease was determined by independent review. Since patients with soft tissue disease were excluded from this trial, Response Evaluation Criteria in Solid Tumors (RECIST)[75] were not applied. Per 2017 INRC, the appearance of any new soft tissue lesions that were also MIBG or FDG-PET avid constituted PD (Supplementary Table 8)[71]. Responses in bone/BM were not required to have confirmatory evaluations, per 2017 INRC (Supplementary Table 8)[71]. Response assessment disaggregated by sex is presented in Supplementary Table 2.

A secondary efficacy endpoint was DoR, defined as time from the first centrally assessed overall response (CR or PR) to PD. Long-term secondary endpoints were PFS and OS. PFS was defined as the time from first infusion of naxitamab to centrally assessed PD or death (whichever came first); OS was defined as the time from the first infusion of naxitamab to death from any cause.

Overall response subgroup analyses were performed by disease status (refractory vs relapsed), baseline disease location (bone vs BM vs bone plus BM), prior anti-GD2 therapy, CS at enrollment, by response compartment, and as an ad hoc analysis by *MYCN* status. Ad hoc analysis of CS change from baseline to best response was summarized and presented in a waterfall plot.

### Safety

Safety was assessed by the type, frequency, severity, seriousness, and duration of AEs after initiation of GM-CSF. AEs were graded for severity according to the National Cancer Institute (NCI) Common Terminology Criteria for Adverse Events (CTCAE) version 4.0 and assessed by trial investigators if they were related to naxitamab or naxitamab plus GM-CSF treatment (hereafter referred to as treatment-related). If an AE grade varied throughout the AE period, the most severe grading observed was assigned. Safety was monitored by an independent data monitoring committee.

To monitor general well-being and patient functional capacity, performance status was assessed at baseline; before Cycles 4, 7, 10, and 13; and at EOT using either the Lansky Play-Performance Scale (for patients age <16 years)[76] or the Karnofsky Performance Status Scale (for patients age ≥16 years)[77]. Pain during infusions was assessed by the FLACC scale for patients ≤5 years old at screening or Wong–Baker FACES scale for patients ≥6 years at screening[60,78].

The number and percentage of infusions administered without the need for overnight hospitalization (i.e., in the outpatient setting) and the number of naxitamab doses administered as planned were also recorded.

### Statistical analysis

All patients who began an infusion of naxitamab were included in the safety population. The efficacy analyses included all patients who began an infusion of naxitamab and who at baseline had centrally assessed, evaluable disease in the bone/BM, were anti-drug antibody-negative, and did not have preplanned radiotherapy for metastatic lesions. Patients with evaluable disease at baseline who could not be evaluated for postbaseline response were considered non-responders. The ORR was defined as the proportion of patients obtaining an overall response of CR or PR (i.e., responders).

Assuming a true ORR of 45%, 37 patients were sufficient in the efficacy population. A lower limit of the 95% exact CI for the ORR of ≥20% was the threshold for determining efficacy. Determination of the trial's total planned sample size was based on extrapolated results from a meta-analysis of COG trials including almost exclusively HR patients (92%), which yielded a PFS at 3 years of 18% with a CI lower limit of 12%[79]. Assuming a 3-year PFS of 23%, a sample size of 80 patients was sufficient to demonstrate an approximately 80% probability that the lower bound of the 2-sided 95% CI for the 3-year PFS is above 12%. With an assumed low attrition rate, the overall inclusion target was set at 85 patients. For response rates (ORR, CR, and PR), 2-sided 95% exact Clopper–Pearson CIs were calculated; DoR, PFS, and OS were estimated using Kaplan–Meier methodology. Statistical analyses were performed using SAS®, version 9.4.

### Reporting summary

Further information on research design is available in the Nature Portfolio Reporting Summary linked to this article.

## Data availability

Y-mAbs is committed to sharing clinical data with qualified external researchers for independent scientific research. Source data are provided with this paper. The redacted protocol and statistical analysis plan are available as online supplementary materials. The underlying data for this study cannot otherwise be made publicly available due to ethical and legal concerns for patient privacy. Researchers may send requests for the aggregate data from this interim analysis between 3 and 24 months after online publication to datasharing@ymabs.com. Access will be provided following review and approval of a research proposal and execution of a data sharing agreement, consistent with local, state, and federal laws and regulations and the rights of study participants. Source data are provided with this paper.

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

## Acknowledgements

The authors wish to thank all the patients, family members, and staff from all the units that participated in the trial. Statistical analysis and data quality-control checks were provided by Per Settergren Sørensen, a former employee of Y-mAbs Therapeutics, Inc. Medical writing and editorial services, under the direction of the authors, were provided by Cynthia D. Gioiello, PharmD, Michelle Jones, PhD, MWC, and Stephen Bublitz, ELS, of MedVal Scientific Information Services, LLC; Kathy Beirne, PhD, and Rosie Morland, PhD, of Excerpta Medica BV; and Sharif Koep, PharmD, and John Lapolla, employees of Y-mAbs Therapeutics, Inc., in the preparation of this manuscript, funded by Y-mAbs Therapeutics, Inc. Funding for the study design, data collection and analysis was provided by Y-mAbs Therapeutics, Inc.

## Author contributions

J.M., G.C.F.C., D.A.M., L.A., K.N., J.F., A.W., M.K.B., A.R.S.S., B.M.L.H., and B.H.K. were involved in collecting and interpreting the data. K.T. and M.D. contributed to the study design, data integrity, statistical analyses, and interpretation of the data. All authors contributed to drafting or critically reviewing the manuscript.

## Competing interests

J.M. has received consulting fees from Y-mAbs Therapeutics, Inc. D.A.M. has been a member of advisory boards/consultant for Clarity Pharmaceuticals, EUSA Pharma, Oncoheroes Biosciences, RayzeBio, Inc., Regeneron, US WorldMeds, and Y-mAbs Therapeutics, Inc.; has received speaker fees from Takeda Israel and Y-mAbs Therapeutics, Inc.; and received travel expenses from AbbVie. L.A. has been a member of an

advisory board for Recordati (EUSA Pharma). K.N. has been a member of advisory boards/consultant for Bayer AG, EUSA Pharma, and Y-mAbs Therapeutics, Inc.; has performed teaching for Bayer AG and Y-mAbs Therapeutics, Inc.; and is a member of a Data Monitoring Committee for Lilly. J.F. has coordinated PI and participation in a compassionate use program for Y-mAbs Therapeutics, Inc. A.W. has been an advisory board member for EUSA Pharma and Y-mAbs Therapeutics, Inc.; and has received expenses for congress attendance from Y-mAbs Therapeutics, Inc. M.K.B. has received consulting fees from Y-mAbs Therapeutics, Inc. A.R.S.S. has been an advisory board member for, and has received expenses for congress attendance from, EUSA Pharma. K.T. and M.D. are employed by, and hold equity with, Y-mAbs Therapeutics, Inc. The remaining authors declare no competing interests.
