## [Transparent Peer Review file · Nature Communications]

The anti-GD2 monoclonal antibody naxitamab plus GM-CSF for relapsed or refractory high-risk neuroblastoma: a phase 2 clinical trial

Corresponding Author: Dr Jaume Mora

Version 0:

Reviewer comments:

Reviewer #1

(Remarks to the Author)

Overview

This very well written manuscript provides the results of an interim analysis that supported the accelerated FDA approval of naxitamab for patients with relapsed/refractory neuroblastoma. There is a rigorously detailed statistical analysis plan, and the analysis has been well conducted according to that plan. However, one concern is that by conducting both an interim and a final analysis using the same two-sided 95% confidence interval, the Type 1 error has been inflated. Another concern is the loss of equipoise in the ongoing trial due to publishing the interim analysis.

Major

In the abstract, it is recommended that the authors

- provide the actual number of responders and the denominator used to calculate the response rate; and,
- provide a standard error or a confidence interval on the point estimates of OS and PFS.

“The primary efficacy endpoint was overall response rate (ORR)...” An endpoint is measured in each patient, whereas ORR is a metric calculated in the aggregated patient cohort. In this case, it seems that the primary endpoint is binary, responder (CR, PR) or non-responder (<PR). The authors are advised to revise the text.

It was good that the study used an independent central review committee to determine the response.

It was good that the authors classified “patients with evaluable disease at baseline who could not be evaluated for response post-baseline” as non-responders, i.e., a conservative definition.

It is good that the trial has a rigorously detailed statistical analysis plan.

The sample size justification in the manuscript mixes the concepts of power and precision. Traditionally, power is presented in conjunction with the statistical test to be used. The efficacy analysis, per the SAP, does not rely on a statistical test, it uses precision: “Efficacy will be concluded if the lower limit of the 95% exact confidence interval for ORR for the FAS cohort 1 exceeds 20%”. The sample size justification would be clearer without the mention of ‘power’. The statistician should be given the opportunity to review/approve the final manuscript wording.

Readers may wonder why the OS timeline goes up to 195 weeks, but the PFS timeline only goes to 78 weeks. Typically, PFS and OS are presented on the same plot. How long were patients planned to be followed onstudy for PFS and OS? (the protocol says 3 years, but this is not in the manuscript.) The following information from the SAP answers these questions, and should be included in the manuscript:

“PFS is defined as the time from the first infusion of naxitamab until PD or death, whichever comes first; data will be censored at the date of last disease evaluation before new anti-NB treatment or last assessment during long-term follow-up, whichever comes first.”

What is the INRGSS (INRG Staging System) stage for these patients? The patients on this study were diagnosed and

staged long after INRGSS became the accepted international standard for disease staging.

Monclair T, Brodeur GM, Ambros PF, Brisse H, Cecchetto G, Holmes K, Kaneko M, London WB, Matthay KK, Nuchtern JG, von Schweinitz D, Cohn SL, Pearson ADJ for the INRG Working Group. The International Neuroblastoma Risk Group (INRG) Staging System. *Journal of Clinical Oncology* 2009 Jan 10;27(2):298-303. Epub 2008 Dec 1. PMID: 19047290
PMCID: PMC2650389.

Information should be provided about the trial's total planned sample size (with sample size justification) and when the trial will end. A very good sample size justification for n=85 was provided in the protocol but not the manuscript.

By conducting both an interim and a final analysis using the same methods (a two-sided 95% confidence interval), the Type 1 error has been inflated. The methods do not utilize an approach for spending Type 1 error. The same 95% confidence intervals are applied in each analysis. The authors should estimate the type 1 error at the end of the study (it will be higher than $\alpha=0.05$). Perhaps it is not too late to amend their statistical plans to make adjustment for multiple testing at the final analysis.

This interim analysis supported the accelerated approval of naxitamab, and thus the justification for performing the analysis. However, by revealing the interim analysis results to the trial investigators and the scientific community, the principle of equipoise has been violated for this trial. It should be noted as a limitation of the final analysis that patient treatment and the enrollment of particular types of patients may be affected due to the loss of equipoise.

Minor

Figure 2c – the Baseline Curie scores are shown in blue. What are the numbers in black? Please provide a definition.

If there were no grade 5 (toxic death) AEs, then the authors should consider saying “grade 3-4” instead of “grade ≥ 3 ”.

Reviewer #2

(Remarks to the Author)

The overall outcome for patients with refractory or relapsed high-risk neuroblastoma is dismal, supporting the study of novel therapies. Immunotherapy approaches, including anti-GD2 antibody therapy have demonstrated improvement in outcome in the upfront therapeutic setting and provided meaningful responses in relapsed disease. The data provide the rationale for the conduct of a Phase II trial of anti-GD2 immunotherapy with Naxitamab. The authors present the feasibility of this approach, associated toxicity, and encouraging observed responses.

The authors are to be congratulated for conducting this trial; it is impressive to see the extent to which therapy was possible in the outpatient setting. It would have been very nice to measure QoL correctly. (there are several validated questionnaires for children with cancer) - I assume that the QoL was quite good in this cohort of patients.

Specific comments include:

Introduction

1. References provided for historical outcomes after relapsed/refractory neuroblastoma may be outdated and should include mention of the more recent trials conducted by SIOPEN (Lode et al., *BJC*, Moreno, et al.; *JCO*, Flaadt et al.; *JCO*) and COG (Mody et al, *Lancet Oncology* and *JCO*).

Methods

2. Reference 13 should be removed as it is not a reference to support the statement in line 97.

3. Minor response is not defined in Suppl. Table 3

Patient population / Table 1

1. As shown here, the patient population "refractory disease" is unusually or not clearly defined. Residual disease with partial response, according to INRC, after induction therapy is not uncommon and is not per se a criterion for refractory disease in COG or SIOPEN high-risk trials. Inadequate response (i.e., refractory disease) in the SIOPEN HR-NBL2 trial is defined by a metastatic response worse than PR and/or a SIOPEN mIBG score >3 ; the authors should use comparable criteria and, if necessary, exclude patients from the analysis if they do not fulfill those. The definition used here impairs the interpretation and should at least be clarified.

2. Why was a patient who did not receive chemotherapy enrolled in the trial (table 1, safety population) although it is defined in the protocol?

3. Time to first relapse is a well-recognized prognostically relevant marker; this information might interest potential readers. I encourage the authors to add this information in Table 1.

4. Although it was not part of the clinical trial, more detailed information about therapy and time before trial enrollment (esp. for the relapsed patients) should be provided, as this information helps to understand ORR and outcomes better.

5. Table 1: Please consider changing "Number of relapses" for better readability. I suggest: 1 relapse: 22 patients, two relapses: 4 patients

Treatment / Efficacy assessment

1. For readers who are not familiar with the protocol, it is not easy to follow the treatment regimen and study-related assessments. Consider adding a figure for clarification.

2. It is not entirely clear what type of imaging was used to assess remission. Was imaging of the metastatic sites sufficient or was whole-body imaging used, as recommended in many studies? Which imaging was mandatory?

3. Were FDG-PET scans mandatory (as defined on p42 in the protocol) or "could" be used for mIBG-negative tumors (line 171). Please clarify.

4. Were bone marrow examinations for minimal disease done, as recommended by the International Neuroblastoma Risk Group Task Force? (Beiske et al and Swerts et al).

5. For easier readability, consider adding CR and PR rates to Table 2

6. Table 3: Was no hematological toxicity observed?

7. Did performance scores or quality-of-life measures (as defined in the trial protocol) change during trial treatment?

Efficacy

1. Please mention the outcomes for patients who have received prior anti-GD2 therapy. This information is of great interest, as a large number of patients who may be considered for Naxitamab therapy in the future have received prior anti-GD2 therapy. The authors should discuss this in "limitations".

Discussion/Conclusion/Limitations

The data presented for this trial represents only a proportion of patients, as patients who progressed prior to entering the trial are not included in the analysis. This introduces a potential bias in the interpretation of outcomes and makes any comparator to historical or recently published data difficult.

In addition, patients with soft tissue lesions were excluded (which I consider reasonable), and the definition of refractory disease is not clear. These limitations leave the impression of a selected patient cohort and that ORRs and outcomes could be overestimated.

It is noteworthy that the outcomes shown here are comparable to the data published by SIOPEN with single agent Dinutuximab beta (Lode 2023, BJC). Although the ORRs in the SIOPEN trial seem to be lower (soft tissue lesions were allowed in this trial), PFS is comparable. Besides, comparable outcomes and ORRs were published in chemo-immunotherapy trials (Moreno 2024, JCO; Mody Lancet Oncology and JCO, Lerman, 2022, JCO), which were used upfront or approaches using anti-GD2 antibodies after allogeneic stem cell transplantation (Flaad 2023, JCO) as consolidation after reinduction for relapsed patients. The discussion section should acknowledge these published data and provide the rationale for using Naxitamab instead of these alternatives. Why is Naxitamab "addressing a significant unmet need..."? (line 364/365).

Reviewer #3

(Remarks to the Author)

This report describes the results of a planned interim analysis of an ongoing multi-center, single arm trial of the anti-GD2 antibody naxitamab given without chemotherapy for patients with relapsed or refractory neuroblastoma limited to bone and bone marrow. The paper is well-written and the results are relevant in the field, although there are issues regarding the generalizability of the results as noted below.

Specific comments

1. Methods, patient population - Could patients have received naxitamab as their first therapy for relapsed disease? The wording implies that this was not first relapse therapy and that the patients included were only those who had already responded to some form of treatment in the relapse setting. Also, by restricting the population to patients who were at least 2 months from time of relapse or PD, the authors have selected for patients with less aggressive disease (ie they have excluded those very challenging to treat patients with galloping neuroblastoma). This needs to be made very clear as it impacts the generalizability of the results.
2. Methods, patient population – The authors state that only patients with disease limited to bone and bone marrow were included. This further impacts the generalizability of the results because not only would patients with new soft tissue lesions at relapse not have been included, but also patients who had residual soft tissue disease at the old primary tumor site measuring >1cm could not have been included because per INRC criteria they would not have been in a CR at the primary soft tissue site. Readers need to be made very aware of the selected nature of the patient population.
3. Methods, patient population (in light of results) – It is notable that per Figure 1, 10 patients were excluded from the efficacy analysis due to lack of baseline disease per independent review and another 5 patients were excluded due to insufficient marrow disease per independent review. This raises concerns about the nature of the eligibility criteria with regard to disease status and suggests that perhaps these criteria were selecting for patients with really minimal disease. $N = 10 + n = 5$ adds up to a substantial portion of the enrolled patient population, and one wonders if there were issues with the eligibility criteria
4. Methods, patient population – The supplemental table with inclusion and exclusion criteria is nice, but some key components of these criteria should be included in the text, as they really impact interpretation of the results. For example, hematologic status is important so that readers will understand the nature of the marrow disease that patients could have had. It is also important to highlight the exclusion criteria re patients with actively progressing disease. It is a little surprising that the presence of antidrug antibody was not an exclusion criterion.
5. Methods, treatment – It is unusual in a clinical trial that duration of therapy for those who remain on study be at the discretion of the treating physician. Readers will want to know why this was the case since it means that the cohort did not receive uniform therapy.
6. Methods, treatment – It is very important for readers (especially patients/parents as well as clinicians who may be considering administering naxitamab to patients) to understand the infrastructure required to safely administer this therapy, since this is really a rate-limiting step in some practices. The supplementary material (Table 2, Figure 1) does spell out the supportive care medications nicely, but it is very important to explicitly state in the manuscript itself what kind of staffing is needed for safety in the administration of a drug can cause hypotension and requires administration of opioids and (at times) ketamine. This is particularly true since the drug is generally given in the outpatient setting and treating teams must have the wherewithal and the right volume to staff for this. It is notable that several centers enrolled only 1 or 2 patients while others enrolled the bulk of the cohort. Could this be related to infrastructure issues as well as to the usual issues of referral patterns and very specific requirements re pattern of disease (eg no soft tissue disease)?
7. Methods, efficacy assessment – Did all sites perform bilateral marrow aspirates/biopsies or did some perform quadrilateral aspirates/biopsies? If any sites performed the latter, which were used for response assessment? Were antibodies targeting synaptophysin the only antibodies permitted as implied by the text?
8. Methods, efficacy assessment – The authors state, "If new soft-tissue lesions appeared post-baseline, these constituted

PD.” They do not state whether there were any size criteria used to declare the presence of a new soft tissue lesion. Please clarify.

9. Results – The objective response rate is certainly of interest, but readers will also want to know how many patients had other categories of response. Please include the number of patients who had PD, SD, MD as relevant to the bone and bone marrow components of the INRC.

10. Results – The median duration of efficacy follow up was 6 months, and the minimum duration was one month. This is very short. The data cut-off was Dec 2021 and it is now 2024, which raises concerns. Were there additional data cut-offs in the time that has elapsed since Dec 2021? One wonders if the duration of response may have been estimable at one of those cut-offs. It is understood that the initiation of additional therapy such as vaccine therapy may have made this difficult, but readers will really want to know whether the effects of naxitamab are sustained or not, and the current data do not address that question well.

11. Results – The same issue regarding timing of the PFS analysis (1-year PFS) arises. Since a fair amount of time has gone by since the time the cohort was enrolled and since Dec 2021, it seems reasonable to evaluate 2 year PFS in this cohort.

12. Results – Figure 2a provides some interesting information regarding response among patients with or without prior anti-GD2 antibody therapy, and the results are a bit surprising. Others have shown that patients who previously received (and may have responded) to anti-GD2 therapy tend to re-respond. However, that does not appear to be the case here as the response rate seems to be higher among patients who had not received prior anti-GD2 antibody therapy. This is not mentioned in the text of the Results section but is interesting. This is mentioned in the Discussion but it is not clear what the authors make of this.

13. Discussion – The authors state that pain was intense but short lived, resolving within 15 minutes. However, data re duration of pain are not included in the Results section of the manuscript or in the supplementary materials. If duration of pain was measured, please add to the supplementary data section. If data regarding duration of pain were not collected, the sentence about this should be modified. The authors note that pain did not lead to treatment discontinuation for any patient. As noted below re Figure 1, it would be helpful to provide the reasons for treatment discontinuation.

14. Discussion – The authors state, “The safety profile of naxitamab is manageable in the outpatient setting.” This requires at least a bit of qualification because sufficient infrastructure and personnel are in fact needed to support this (see comment 6 above). Also, none of the data in the rest of this paragraph are given in the Results section or in the supplementary materials. Please provide the data at least in the supplement.

15. Discussion – The concluding paragraph should be modified in light of the points made above.

Figures/tables

1. Figure 1 – What were the “other” reasons that patients were excluded in addition to not meeting eligibility criteria?

2. Figure 1 – What were the specific adverse events that led to discontinuation of treatment?

3. Table 1 – This study was conducted in the modern era, so stage should be given using INRG rather than INSS.

4. Table 1 – Was INPC status formally assessed in all patients? This system is used routinely in North America however degree of differentiation is sometimes used in other countries to evaluate histology without the added elements of age and MKI included in INPC. Please confirm.

5. Table 1 – The data re number of relapses are confusing. A quick look at the table suggests that the overwhelming majority of patients with relapsed disease were treated on this study in the first relapse setting. However footnote b says that 26 and 37 patients had relapsed at least once before enrollment. One would then think that these patients would not be considered to have been treated at time of first relapse. Please clarify. If these patients were indeed treated in first relapse, this should be added to the limitations section, especially in light of the data regarding the difference in response rates for patients who had received prior anti-GD2 antibody therapy vs those who had not. If the patients had had one prior relapse, then the heading in this row should be “Number of prior relapses” rather than “Number of relapses.”

6. Table 2 – It will be easier for readers to think through this important table if all response are included and not just responses other than CR and PR.

Reviewer #4

(Remarks to the Author)

The authors present the interim findings of study 201 which is a global, single arm, open label phase 2 study evaluating the combination of naxitamab and GMCSF for the treatment of patients with relapsed or refractory neuroblastoma with disease in the bone, bone marrow or both. I appreciate the authors presenting this data but feel that some changes are necessary. Below are the specifics of my review.

Abstract: Well written however, would consider changing the order presented in the results section to present the toxicity data followed by the response data. If this change is taken into consideration, recommend changing this order in the protocol “results” section as well. Additionally, the response data, as written in the results section of the abstract, is confusing. Would delete some of the response data in the abstract (such as the complete response information) and include more data on the toxicity. As an example, 62% of your patients had Grade 3 or 4 hypotension, but you do not include the number. The authors should include those numbers for the hypotension, pain and urticaria in the abstract.

Methods: The authors should state how a “bone lesion” was classified as both bone marrow and bone disease can be MIBG avid.

Statistical Analysis: The authors should report why the patients with anti-drug antibody negative were excluded from the efficacy data. There is conflicting data about whether or not this effects response and thus these patients should not be excluded from the efficacy analysis.

Results: I would recommend presenting toxicity data prior to response data.

Throughout the results, the authors used phrases like “CR rates were numerically higher” and others when describing data. Many of the presented findings were not statistically significant. Thus, those phrases should be removed. Instead the authors

can just state the facts.

For the safety data, Naxitimab is not an easy medication to administer or for patients to receive. Thus, the authors need to provide more information as a supplemental table regarding the supportive care received by the patients. They currently have a figure with the supportive care possibilities. However, they supply no further information about what patients actually received. In the setting of some significant grade 3/4 toxicities, the community would benefit from more details. The authors also included duration (in hours) of certain Grade >3 toxicities. This is not something that is typically reported and should be removed. If the author is trying to make a point they can add that point to the Discussion Section.

I think that it is very interesting that of the 50 SAEs, only 62% were considered related. Many of the side effects related to this therapy can also be related to supportive care and thus there must be overlap. The authors should include information regarding the SAEs that were “unrelated”. Again, in this section the authors use phrase like, “appeared to reduce” when describing the frequency of toxicities. Unless it is statistically significant, the authors should refrain from those sayings. Discussion: The discussion section is lacking discussion regarding other anti-GD2 therapies and how this treatment may compare. Further it is drawing conclusion (such as “response to naxitimab was similar across subgroups including MYCN amplification”) when these findings were not statistically significant AND the patient population had an unexpectedly low number of patients with MYCN amplification.

The discussion section is lacking further discussion about the observed toxicities. It merely restates the results and does not elaborate. I think that it is prudent for the authors to discuss in further detail the 62% of patients with Grade 3 or greater hypotension and discuss some of the other results, not just restate them.

In general I think the tone of the manuscript and statements made throughout should be toned down and stick to the data and facts.

Figures/Tables:

Figure 1.

In figure 1, I am surprised that 10% of patients did not meet eligibility criteria and were excluded. This feels high. Additionally I have significant concerns that 10 patients did not have baseline disease evaluated per independent reviewer and another 5 did not have baseline bone marrow disease per independent review. That equals 20% of the analyzed patients included in the cohort. While I am pleased that the investigators removed them from the efficacy group, they did not discuss this in the manuscript and it makes me question other aspects of the data. This must be discussed in the body of the manuscript as the number is very high and technically that would mean that the patients should not have enrolled on the study.

Figure 2.

Letter a figure should be removed and the data should be included in Table 2. Letter b data needs revision. It is impossible to see the overlying colors (relapsed/refractory and response). Further many of the lanes on the swimmers plots do not make sense and need to be double checked. Response End circles should just be removed as the overlay between that and response is difficult. As such, it is hard to see the progressive disease boxes.

Figure 3.

I would remove the OS Kaplan Meier curve as many patients went on to receive other therapy.

Table 1. MYCN is under-represented in this R/R patient population at only 14%. Thus, the authors should discuss this in the body of the manuscript.

Table 3.

When reviewing the supplementary table and this table I think that a few changes should be made. Recommend including the Grade 4 anaphylaxis and the Grade 4 respiratory depression in the supplementary table in Table 3 instead.

Version 1:

Reviewer comments:

Reviewer #1

(Remarks to the Author)

The authors have satisfactorily responded to the review.

Reviewer #2

(Remarks to the Author)

The authors responded well to the reviewers' comments. I think it is very important for the field that the results of this trial are published. Even though I still feel that the results are overestimated due to a highly selected cohort of patients, I think the manuscript is ready for publication, and I have no further comments.

Reviewer #4

(Remarks to the Author)

1. The abstract, as written, is impossible to follow. For example, it reads “among 26 responders” followed up by (CR+PR ; n=52)...this does not make sense unless you read the manuscript and realize that 22 patients were excluded from the response analysis. In the abstract, the authors need to do a better job describing the results. It is too confusing as written. I believe a sentence in the abstract stating that 22 patients were excluded from the efficacy analysis will make this more clear.
2. While the study evaluates an important drug and there are encouraging results for a specific patient population, I feel that the trial and interim analysis as presented does not meet the scientific rigor and requirements expected from studies published in Nature Communications. There are too many risks of potential biasing of data including the enrollment criteria and definition and number of patients excluded from the efficacy data. Additionally, the lack of transparency around some of the toxicity data leads to questions. I appreciate the revisions and feel that the revised manuscript is improved. However, I

think the number of inherent flaws in the clinical trial design and included data are concerning unless further addressed.

3. In the limitations section, more time must be focused on the inclusion criteria of this study in that it does not represent many of the patients one thinks of when considering a relapsed or refractory NB patient. The average reader, lacking neuroblastoma expertise, may not be able to distinguish that the studied patient population is very specific and represents a small number of R/R patients. For example, it is worth noting in the manuscript that this patient population likely represents a skewed patient population from prior reported clinical trials. This is evidenced in the baseline Lansky and Karnofsky scores being 100% and 96%. This likely is a reflection of minimal disease present at baseline and must be addressed in the limitations section.

4. While the language in the discussion section has been toned down, I still feel that the results are overstated and that the authors did not adequately address the limitations and potential bias of the results. With this patient population, it is very difficult to make sweeping generalizations about comparison to other studies.

6. I still have significant concerns about the patients that were excluded (15 of 74 patients) from the efficacy data. The information provided in the authors response has heightened my concerns with bias and I think this needs to be further addressed.

Version 2:

Reviewer comments:

Reviewer #4

(Remarks to the Author)

I believe the authors have appropriately addressed the prior concerns in the revised manuscript.

REVIEWER COMMENTS

Reviewer #1 (Remarks to the Author): with expertise in biostatistics, clinical trials, neuroblastoma

Overview

This very well written manuscript provides the results of an interim analysis that supported the accelerated FDA approval of naxitamab for patients with relapsed/refractory neuroblastoma. There is a rigorously detailed statistical analysis plan, and the analysis has been well conducted according to that plan. However, one concern is that by conducting both an interim and a final analysis using the same two-sided 95% confidence interval, the Type 1 error has been inflated. Another concern is the loss of equipoise in the ongoing trial due to publishing the interim analysis.

REPLY: Thank you for raising this point. We acknowledge that this report of results from a prespecified interim analysis risks the loss of equipoise that may affect ongoing patient enrollment and treatment. As now noted in the Limitations section, these risks are perhaps best measured against the need for sharing the data with the clinical community in a peer-reviewed forum (page 19).

With respect to the potential multiplicity issue, the interim reporting of the ORR was performed with sufficient precision to conceptually qualify as final in the sense that any subsequent analysis of this endpoint should be considered confirmatory of the overall hypothesis. A standard multiplicity adjustment in this context would be a Bonferroni-type scaling of P values. Given that we see $P < 0.0001$, the current result would be supported. At this point, with the result of the interim analysis known, we do not think that adjusting the statistical plan would be appropriate. In the final reporting, we will point out that no formal adjustment for multiplicity has been performed. We also note (page 19) that the final trial analysis will include a larger patient population and will provide efficacy and safety data on long-term outcomes, including overall survival (OS).

Major

In the abstract, it is recommended that the authors

- provide the actual number of responders and the denominator used to calculate the response rate; and,
- provide a standard error or a confidence interval on the point estimates of OS and PFS.

REPLY: We have updated the Abstract to include the actual number of responders, the denominator used to calculate the response rate, and the CI for reported OS and progression-free survival (PFS; page 4).

“The primary efficacy endpoint was overall response rate (ORR)...” An endpoint is measured in each patient, whereas ORR is a metric calculated in the aggregated patient cohort. In this case, it seems that the primary endpoint is binary, responder (CR, PR) or non-responder (<PR). The authors are advised to revise the text.

REPLY: We have updated the Methods in the Abstract (page 4) and manuscript body (page 8) to clarify that the primary endpoint was overall response, with responders defined as those achieving CR or partial response (PR; pages 4 and 8). Similarly, we define the ORR as the proportion of patients obtaining an overall response of CR or PR (page 10).

It was good that the study used an independent central review committee to determine the response.

It was good that the authors classified “patients with evaluable disease at baseline who could not be evaluated for response post-baseline” as non-responders, i.e., a conservative definition.

It is good that the trial has a rigorously detailed statistical analysis plan.

REPLY: We appreciate your comments and are pleased to be aligned on these important matters.

The sample size justification in the manuscript mixes the concepts of power and precision. Traditionally, power is presented in conjunction with the statistical test to be used. The efficacy analysis, per the SAP, does not rely on a statistical test, it uses precision: “Efficacy will be concluded if the lower limit of the 95% exact confidence interval for ORR for the FAS cohort 1 exceeds 20%”. The sample size justification would be clearer without the mention of ‘power’. The statistician should be given the opportunity to review/approve the final manuscript wording.

REPLY: Thank you for your comment. The statements addressing sample size justification and efficacy threshold have been revised accordingly and approved by the statistician (page 11).

Readers may wonder why the OS timeline goes up to 195 weeks, but the PFS timeline only goes to 78 weeks. Typically, PFS and OS are presented on the same plot. How long were patients planned to be followed on study for PFS and OS? (The protocol says 3 years, but this is not in the manuscript.) The following information from the SAP answers these questions, and should be included in the manuscript:

“PFS is defined as the time from the first infusion of naxitamab until PD or death, whichever comes first; data will be censored at the date of last disease evaluation before new anti-NB treatment or last assessment during long-term follow-up, whichever comes first.”

REPLY: Thank you for your comment. The planned follow-up for the final trial is 3 years for both OS and PFS, which we now note on page 8 of the manuscript. The Kaplan-Meier curves present the follow-up for OS and PFS at the time of data cutoff for the interim analysis. The follow-up for PFS is shorter as more censoring rules apply to this endpoint. Definitions for PFS and OS and description of the data censoring procedures are now included in the legends for Figure 3 showing the Kaplan-Meier curves (page 36).

What is the INRGSS (INRG Staging System) stage for these patients? The patients on this study were diagnosed and staged long after INRGSS became the accepted international standard for disease staging.

REPLY: Thank you for this question. While we recognize that the International Neuroblastoma Risk Group Staging System (INRGSS) is the current accepted standard for disease staging, this study was designed for regulatory purposes and to replicate the phase 1/2 trial (ISS 12-230) conducted at Memorial Sloan Kettering and initiated in 2012 (Kushner et al., 2018), which utilized the International Neuroblastoma Staging System (INSS). Unfortunately, it is not possible to change the study design in the protocol at this time. Importantly, INSS staging remains clinically relevant and is still currently used in clinical practice.

It is worth noting that in the Trial 201 interim analysis, 89% of patients in both the safety and efficacy populations had INSS stage 4 disease, which is equivalent to INRG stage M (Monclair et al., 2009). Additionally, the Trial 201 inclusion criterion for HR disease at the time of diagnosis is as per INRG (Cohn et al., 2009).

Information should be provided about the trial's total planned sample size (with sample size justification) and when the trial will end. A very good sample size justification for n=85 was provided in the protocol but not the manuscript.

REPLY: Thank you for your suggestion. Justification for the trial's total planned sample size has been added to the Methods section of the manuscript (page 11). End of trial was defined as the last subject's last visit in the long-term follow-up period, and this information is now in the Methods section (page 8).

By conducting both an interim and a final analysis using the same methods (a two-sided 95% confidence interval), the Type 1 error has been inflated. The methods do not utilize an approach for spending Type 1 error. The same 95% confidence intervals are applied in each analysis. The authors should estimate the type 1 error at the end of the study (it will be higher than $\alpha=0.05$). Perhaps it is not too late to amend their statistical plans to make adjustment for multiple testing at the final analysis.

REPLY: Thank you for your suggested edits; however, we do not think that adjusting the statistical plan would be appropriate at this point. As mentioned above, the current interim result for ORR should be considered final and any subsequent reporting of ORR in the final analysis should be viewed as confirmatory. In the final reporting, we will point out that no formal adjustment for multiplicity was performed.

This interim analysis supported the accelerated approval of naxitamab, and thus the justification for performing the analysis. However, by revealing the interim analysis results to the trial investigators and the scientific community, the principle of equipoise has been violated for this trial. It should be noted as a limitation of the final analysis that patient treatment and the enrollment of particular types of patients may be affected due to the loss of equipoise.

REPLY: Thank you for your comment. In the Limitations section (page 19), we acknowledge that the potential loss of equipoise may have an impact on ongoing patient enrollment and treatment, which is an important consideration that we measure against the need for sharing the data with clinical community.

Minor

Figure 2c – the Baseline Curie scores are shown in blue. What are the numbers in black? Please provide a definition.

REPLY: Thank you for your question. The numbers in black represent the maximum change in Curie Score (CS). The figure legend has been updated to provide this information (page 34).

If there were no grade 5 (toxic death) AEs, then the authors should consider saying “grade 3-4” instead of “grade ≥ 3 ”.

REPLY: Thank you for this suggestion; we have changed all instances of “Grade ≥ 3 ” to “Grade 3-4.”

Reviewer #2 (Remarks to the Author): with expertise in neuroblastoma, therapy

The overall outcome for patients with refractory or relapsed high-risk neuroblastoma is dismal, supporting the study of novel therapies. Immunotherapy approaches, including anti-GD2 antibody therapy have demonstrated improvement in outcome in the upfront therapeutic setting and provided meaningful responses in relapsed disease. The data provide the rationale for the conduct of a Phase II trial of anti-GD2 immunotherapy with Naxitamab. The authors present the

feasibility of this approach, associated toxicity, and encouraging observed responses. The authors are to be congratulated for conducting this trial; it is impressive to see the extent to which therapy was possible in the outpatient setting. It would have been very nice to measure QoL correctly. (there are several validated questionnaires for children with cancer) - I assume that the QoL was quite good in this cohort of patients.

REPLY: We agree that quality of life (QoL) is a particularly important consideration. We appreciate the comment on the availability of validated questionnaires for children with cancer, while recognizing the unique challenges of patients with HR neuroblastoma. There is currently no validated, neuroblastoma-specific QoL tool available for use during a cycle of naxitamab treatment.

In Trial 201, we have used a simple, intuitive non-validated visual analog scale focusing on activity and happiness as a proxy for assessing QoL. These scales were implemented in Protocol 201 version 11.0 (dated 03-Dec-2020). The data were, however, considered immature for reporting at the time of this prespecified interim analysis.

Specific comments include:

Introduction

1. References provided for historical outcomes after relapsed/refractory neuroblastoma may be outdated and should include mention of the more recent trials conducted by SIOOPEN (Lode et al., BJC, Moreno, et al.; JCO, Flaadt et al.; JCO) and COG (Mody et al, Lancet Oncology and JCO).

REPLY: Thank you for recommending these seminal studies. The following references have been added to the manuscript: SIOOPEN (Flaadt et al., 2023; Lode et al., 2023; Moreno et al., 2024) and COG (Mody et al., 2017; Mody et al., 2020).

Methods

2. Reference 13 should be removed as it is not a reference to support the statement in line 97.

REPLY: We have removed the reference in line 97.

3. Minor response is not defined in Suppl. Table 3

REPLY: Thank you for your comment. Supplementary Table 4 defines tumor response at metastatic bone sites and bone marrow (BM) metastatic response according to INRC (Park et al., 2017). Minor response is used when defining overall response and is therefore not applicable to

Supplementary Table 4. We have added Supplementary Table 5 that includes the determination of overall response (including minor response) per INRC.

Patient population / Table 1

1. As shown here, the patient population "refractory disease" is unusually or not clearly defined. Residual disease with partial response, according to INRC, after induction therapy is not uncommon and is not per se a criterion for refractory disease in COG or SIOOPEN high-risk trials. Inadequate response (i.e., refractory disease) in the SIOOPEN HR-NBL2 trial is defined by a metastatic response worse than PR and/or a SIOOPEN mIBG score >3; the authors should use comparable criteria and, if necessary, exclude patients from the analysis if they do not fulfill those. The definition used here impairs the interpretation and should at least be clarified.

REPLY: Thank you for this important comment. We agree that the reviewer's request to use criteria comparable to those employed by SIOOPEN HR trials and to exclude patients who do not fulfill these criteria would address an important literature gap. Since this study was designed to follow the Phase 1/2 trial (Cheung et al., 2017; Kushner et al., 2018), it does not utilize SIOOPEN criteria. The analysis proposed by the reviewer would substantively deviate from the Trial 201 protocol and is therefore not feasible. We have added this limitation to the manuscript (page 19).

Importantly, and as now addressed in the Limitations section, a universal definition of refractory disease does not currently exist (page 20). However, the results of this prespecified interim analysis provide additional insights into patients who do not achieve a CR at the end of induction, which can supplement the growing evidence base supporting a clinically meaningful definition of refractory disease.

2. Why was a patient who did not receive chemotherapy enrolled in the trial (table 1, safety population) although it is defined in the protocol?

REPLY: Thank you for your question. This particular patient received dinutuximab in combination with irinotecan and temozolomide prior to enrollment, which was recorded in the database as immunotherapy. As a result, recording of prior chemotherapy was missed. This error will be corrected during final data analysis, pending completion of the trial.

3. Time to first relapse is a well-recognized prognostically relevant marker; this information might interest potential readers. I encourage the authors to add this information in Table 1.

REPLY: Thank you for this helpful recommendation. While we agree that the first relapse is a well-recognized prognostically relevant marker, the date of the CR to first relapse is not available.

4. Although it was not part of the clinical trial, more detailed information about therapy and time before trial enrollment (esp. for the relapsed patients) should be provided, as this information helps to understand ORR and outcomes better.

REPLY: Thank you for this suggestion. We agree that more detailed information about therapy and the time before trial enrollment would be useful. Although the trial is ongoing, with data collection still in progress, we have added the median time from initial diagnosis to first relapse, the median time from first relapse to second relapse, and the median time from last relapse to trial entry to Table 1 (page 38). To ensure the most accurate and complete data, a list of therapies prior to enrollment will be provided when reporting the final analysis.

5. Table 1: Please consider changing "Number of relapses" for better readability. I suggest: 1 relapse: 22 patients, two relapses: 4 patients

REPLY: Thank you for your suggestion. The table subheading has been clarified (page 38).

Treatment / Efficacy assessment

1. For readers who are not familiar with the protocol, it is not easy to follow the treatment regimen and study-related assessments. Consider adding a figure for clarification.

REPLY: Thank you for your suggestion. Schematics of the treatment cycles and study-related assessments have been added to the Supplementary material (Supplementary Figure 1a and 1b, pages 15 and 16).

2. It is not entirely clear what type of imaging was used to assess remission. Was imaging of the metastatic sites sufficient or was whole-body imaging used, as recommended in many studies? Which imaging was mandatory?

REPLY: Whole-body meta-iodobenzylguanidine (MIBG) scans were mandatory for all patients. Anatomical scans covered the pelvis, abdomen, and chest. Imaging modalities used at every response assessment included computed tomography (CT) or magnetic resonance imaging (MRI) and MIBG scan or 18F-Fluorodeoxyglucose Positron Emission Tomography (FDG-PET [if not MIBG-avid]). Alternatively, MIBG-SPECT/CT could be used.

3. Were FDG-PET scans mandatory (as defined on p42 in the protocol) or "could" be used for mIBG-negative tumors (line 171). Please clarify.

REPLY: Thank you for your question. For MIBG-non-avid tumors, FDG-PET scans were mandatory. This is stated in the Protocol in footnote 7 associated with Table in Section 3

(Flowchart) and in Section 9.1.1 addressing “response.” In the manuscript, use of FDG-PET in cases of MIBG-non-avid neuroblastoma is noted in the Methods section (page 9).

4. Were bone marrow examinations for minimal disease done, as recommended by the International Neuroblastoma Risk Group Task Force? (Beiske et al and Swerts et al).

REPLY: BM examinations for minimal disease were not performed.

5. For easier readability, consider adding CR and PR rates to Table 2

REPLY: Thank you for this suggestion. CR and PR rates have been added to Table 2 where applicable (pages 40 and 41).

6. Table 3: Was no hematological toxicity observed?

REPLY: Thank you for your question. As reported in the manuscript, there were no treatment-related hematological toxicities that occurred in $\geq 10\%$ of patients. Please see below for additional hematological toxicity data (not shown in the manuscript) for Grade 3-4 events, all events and those related to naxitamab + GM-CSF.

	Patients (N=74) n (%)
Grade 3-4 blood and lymphatic system disorders in $\geq 2\%$ of subjects	
Neutropenia	8 (11)
Lymphopenia	4 (5)
Anemia	3 (4)
Thrombocytopenia	2 (3)
Febrile neutropenia	2 (3)
Leukopenia	2 (3)
Grade 3-4 blood and lymphatic system disorders related to naxitamab + GM-CSF by SOC and PT	
Thrombocytopenia	2 (3)
Anemia	1 (1)
Neutropenia	1 (1)

7. Did performance scores or quality-of-life measures (as defined in the trial protocol) change during trial treatment?

REPLY: Thank you for your question. The mean Lansky Score (Lansky et al., 1987), used for children <16 years of age (n=71) was 96.6 at baseline and 95.1 at end of treatment. The mean

absolute change in Lansky Score from baseline to EOT was -1.9 (median change, 0; range, -30 to 20).

Three patients had performance score assessed by the Karnofsky Score (for children ≥ 16 years of age) (Karnofsky and Burchenal, 1949). There was no change in score from baseline for these patients. The mean Karnofsky Score was 100 at both baseline and at end of treatment.

The QoL data, as noted above, were considered immature for reporting at the time of this prespecified interim analysis.

Efficacy

1. Please mention the outcomes for patients who have received prior anti-GD2 therapy. This information is of great interest, as a large number of patients who may be considered for Naxitamab therapy in the future have received prior anti-GD2 therapy. The authors should discuss this in "limitations".

REPLY: Thank you for your comments. We agree that outcomes in patients who have received prior anti-GD2 therapy are of significant interest and are presented in the Results section (page 12), Table 2 (page 42), and further discussed on page 16 of the manuscript. Briefly, 12 of the 13 patients with prior exposure to anti-GD2 therapy had relapsed disease and failed to achieve CR prior to enrolling in Trial 201 and receiving single-agent anti-GD2 therapy. Patients with relapsed disease are generally very sick, are heavily pretreated, and have a poor prognosis (DuBois et al., ASCO, 2022; Moreno et al., 2017).

We note in the Discussion section that this study was not powered for subgroup analyses, including prior anti-GD2 therapy, which therefore limits any firm conclusions. Nevertheless, because little is known about the response to anti-GD2 therapy in this specific patient population, we believe these data will make an important contribution to the published literature.

Discussion/Conclusion/Limitations

The data presented for this trial represents only a proportion of patients, as patients who progressed prior to entering the trial are not included in the analysis. This introduces a potential bias in the interpretation of outcomes and makes any comparator to historical or recently published data difficult.

REPLY: Thank you for your comment. The 2-month cutoff from the time of relapse or PD allowed time for patients to receive and respond to therapy prior to study enrollment and thereby distinguished patients who were refractory to treatment from those who had PD or active relapse. We have clarified this rationale in the Methods section (page 6). By clearly demarcating between refractory disease and PD or active relapse, the inclusion/exclusion criteria provide an important

perspective on these still-evolving definitions. We agree that this is an important consideration for any comparison with published studies using different inclusion criteria.

In addition, patients with soft tissue lesions were excluded (which I consider reasonable), and the definition of refractory disease is not clear. These limitations leave the impression of a selected patient cohort and that ORRs and outcomes could be overestimated.

REPLY: Thank you for your comment. We agree that single use of naxitamab for the treatment of soft tissue lesions is not considered appropriate, which is why these patients were excluded from the trial; this is now listed as a limitation on page 19. As noted above, we acknowledge that there is no universal definition of “refractory” disease, and we provide additional context on this important issue in the Limitations section (page 20).

It is noteworthy that the outcomes shown here are comparable to the data published by SIOOPEN with single agent Dinutuximab beta (Lode 2023, BJC). Although the ORRs in the SIOOPEN trial seem to be lower (soft tissue lesions were allowed in this trial), PFS is comparable. Besides, comparable outcomes and ORRs were published in chemo-immunotherapy trials (Moreno 2024, JCO; Mody Lancet Oncology and JCO, Lerman, 2022, JCO), which were used upfront or approaches using anti-GD2 antibodies after allogeneic stem cell transplantation (Flaad 2023, JCO) as consolidation after reinduction for relapsed patients. The discussion section should acknowledge these published data and provide the rationale for using Naxitamab instead of these alternatives. Why is Naxitamab "addressing a significant unmet need..."? (line 364/365).

REPLY: Thank you for your comment. We have updated the Discussion section to reference these published data. Direct comparisons between naxitamab and other anti-GD2 monoclonal antibodies (mAbs) will require additional studies, including head-to-head trials, the need for which is mentioned in the Discussion section (page 16).

We also discuss how naxitamab addresses a significant unmet need. Specifically, we note that the interim results uniquely demonstrate the safety and efficacy of anti-GD2 mAb as single-agent therapy (in combination with GM-CSF) in a patient population with residual disease in bone and/or BM. Many of these patients, as noted above, have been heavily pretreated for relapsed disease and have a dismal prognosis. These findings therefore contribute to the broader body of research evaluating various combination regimens, including chemotherapy, chemoimmunotherapy, and single-agent immunotherapy, for diverse and vulnerable patients.

Reviewer #3 (Remarks to the Author): with expertise in neuroblastoma, therapy

This report describes the results of a planned interim analysis of an ongoing multi-center, single arm trial of the anti-GD2 antibody naxitamab given without chemotherapy for patients with

relapsed or refractory neuroblastoma limited to bone and bone marrow. The paper is well-written and the results are relevant in the field, although there are issues regarding the generalizability of the results as noted below.

Thank you for your kind comments.

Specific comments

1. Methods, patient population - Could patients have received naxitamab as their first therapy for relapsed disease? The wording implies that this was not first relapse therapy and that the patients included were only those who had already responded to some form of treatment in the relapse setting. Also, by restricting the population to patients who were at least 2 months from time of relapse or PD, the authors have selected for patients with less aggressive disease (ie they have excluded those very challenging to treat patients with galloping neuroblastoma). This needs to be made very clear as it impacts the generalizability of the results.

REPLY: Thank you for your question and comments. We have clarified in the Methods section on page 6 and in the footnote of Table 2 (page 38) of the manuscript that relapsed patients were defined as those who have failed “salvage” therapy administered for the relapse or PD prior to enrollment. Therefore, naxitamab could not have been the first therapy for PD or relapsed disease. We have noted in the Methods section (page 7) that the 2-month cutoff from the time of relapse or PD was established to distinguish patients who were refractory to treatment from those who had active PD or relapse. We acknowledge that this is an important consideration for any comparison with published studies using different inclusion criteria.

The study included patients with relapsed or refractory (R/R) neuroblastoma in the bone and/or BM compartments only, where anti-GD2 therapy alone was thought to be most effective, and we believe that this report will help clarify the treatment responsiveness in this patient population. Notably, anti-GD2 mAbs alone are not utilized as initial treatment for new PD and are not appropriate for soft tissue disease. Patients with soft tissue disease or active PD, who were excluded from this trial, may benefit from chemotherapy or chemotherapy combined with anti-GD2 mAbs. Clarifying the impact of anti-GD2 therapy for these and other subpopulations, such as the one you reference, will require additional studies.

At present, this single-arm study includes patients with disease in locations that may respond to naxitamab alone (with GM-CSF) and thereby permits evaluation of naxitamab treatment responses without the confounding effects of chemotherapy regimens that would have otherwise been required for those with active PD or soft tissue disease at baseline.

We now acknowledge in the Limitations section (page 19), that Trial 201 excluded patients with soft-tissue disease or active PD and further note that these considerations limit the generalizability of the findings across a diverse patient population (pages 16 and 19).

2. Methods, patient population – The authors state that only patients with disease limited to bone and bone marrow were included. This further impacts the generalizability of the results because not only would patients with new soft tissue lesions at relapse not have been included, but also patients who had residual soft tissue disease at the old primary tumor site measuring >1cm could not have been included because per INRC criteria they would not have been in a CR at the primary soft tissue site. Readers need to be made very aware of the selected nature of the patient population.

REPLY: Thank you for pointing this out. We have added details to the Methods (pages 6-9) to provide clarity on the patient population and, as mentioned above, acknowledge in the Limitations section that patients with residual soft-tissue disease after salvage therapy or active PD were excluded from the trial (page 19). Patients with residual soft-tissue lesions that responded to salvage therapy and who had residual bone/BM disease at baseline were permitted to enroll in Trial 201.

Per 2017 INRC, the appearance of any new soft tissue lesions that were also MIBG or FDG-PET avid constituted PD in the trial. As noted, the inclusion and exclusion criteria limited the patient population to those with disease in the bone and BM only, a population in which single drug therapy was thought to be most effective. We acknowledge that this is an important consideration for any comparison with published studies using different inclusion criteria.

3. Methods, patient population (in light of results) – It is notable that per Figure 1, 10 patients were excluded from the efficacy analysis due to lack of baseline disease per independent review and another 5 patients were excluded due to insufficient marrow disease per independent review. This raises concerns about the nature of the eligibility criteria with regard to disease status and suggests that perhaps these criteria were selecting for patients with really minimal disease. $N=10 + n=5$ adds up to a substantial portion of the enrolled patient population, and one wonders if there were issues with the eligibility criteria

REPLY: Thank you for your comments, and we understand your concern, which was also raised by another reviewer. We have added additional details to the footnote of Figure 1 (page 33) and discuss the selective nature of the population in the Limitations section (page 19). Please note that during the trial, the protocol was amended to prevent enrollment of patients with very limited disease burden (BM infiltration of $\leq 5\%$).

Of the 74 patients in the efficacy population, 22 were excluded for the following reasons:

- 5 patients had BM disease only with a BM infiltration of $\leq 5\%$. The requirement of “BM infiltration $>5\%$ for patients with “BM disease only” became an inclusion criterion during the conduct of Trial 201. The reason for adding this requirement was to exclude patients with very limited disease burden. Accordingly, patients with “BM disease only and BM

infiltration of $\leq 5\%$ ” enrolled according to older versions of the protocol and were excluded from the efficacy population.

- 2 patients had preplanned radiotherapy to target metastatic sites. In accordance with the protocol, they were allowed to enroll but were excluded from the efficacy population.
- 4 patients had positive anti-drug antibody (ADA) titers at baseline and were excluded from the efficacy population as per protocol.
- 1 patient had no postbaseline response assessment prior to data cutoff and was excluded from the efficacy analysis population.
- 10 patients had no baseline evaluable disease as per independent review. These patients all had a CS of 0 in bone:
 - In 2 patients, the 2 reviewers agreed to a CS of 0.
 - In 1 patient, the 2 reviewers agreed that the patient had a CS of 1 in retroperitoneal soft tissue and not in spine. The patient was excluded, as only residual disease in bone and/or BM was allowed per protocol. Of note, for this patient, 1 reviewer also found a CS of 1 in the bones. This was not seen by the other reviewer. Adjudication by the third reviewer confirmed a CS of 0 in bones.
 - In 7 patients, the CS was 0 per 1 reviewer, but 1-2 per the other reviewer (2 patients with CS of 2). Adjudication by a third reviewer confirmed a CS of 0.
 - In 3 patients, no BM slides were sent for independent review.

4. Methods, patient population – The supplemental table with inclusion and exclusion criteria is nice, but some key components of these criteria should be included in the text, as they really impact interpretation of the results. For example, hematologic status is important so that readers will understand the nature of the marrow disease that patients could have had. It is also important to highlight the exclusion criteria re patients with actively progressing disease. It is a little surprising that the presence of antidrug antibody was not an exclusion criterion.

REPLY: Thank you for your suggestion. Highlights of important inclusion and exclusion criteria, including hematological status, have been added to the manuscript (page 7). Presence of ADA was not listed as a specific item in the exclusion criteria because the assay had not been validated for use at the time of trial design; however, those who were ADA positive at baseline, as determined retrospectively, were excluded from the efficacy analysis.

5. Methods, treatment – It is unusual in a clinical trial that duration of therapy for those who remain on study be at the discretion of the treating physician. Readers will want to know why this was the case since it means that the cohort did not receive uniform therapy.

REPLY: The duration of therapy varied because it was dependent on patient response. Treatment criteria were consistent in terms of the primary endpoint (CR or PR), centrally assessed according to the 2017 INRC. Treatment cycles were repeated every 4 weeks until CR or PR followed by 5 additional cycles every 4 weeks. Any subsequent cycles were repeated every 8

weeks through 101 weeks from the first infusion at the discretion of the investigator.

6. Methods, treatment – It is very important for readers (especially patients/parents as well as clinicians who may be considering administering naxitamab to patients) to understand the infrastructure required to safely administer this therapy, since this is really a rate-limiting step in some practices. The supplementary material (Table 2, Figure 1) does spell out the supportive care medications nicely, but it is very important to explicitly state in the manuscript itself what kind of staffing is needed for safety in the administration of a drug can cause hypotension and requires administration of opioids and (at times) ketamine. This is particularly true since the drug is generally given in the outpatient setting and treating teams must have the wherewithal and the right volume to staff for this. It is notable that several centers enrolled only 1 or 2 patients while others enrolled the bulk of the cohort. Could this be related to infrastructure issues as well as to the usual issues of referral patterns and very specific requirements re pattern of disease (eg no soft tissue disease)?

REPLY: Thank you for your comment. We acknowledge that successful administration of naxitamab requires a coordinated and prepared multidisciplinary team. We have noted in the manuscript that site initiation meetings were conducted to ensure the sharing of best practices for safe and effective administration of infusion, pre-medications, and supportive medications according to each site’s established processes and procedures (page 8). Single-site experience with successful outpatient administration of naxitamab has recently been published (Trovilleon et al., 2024), and we have included a reference to the paper.

It should be noted that neuroblastoma is a rare disease, which resulted in low enrollment at some sites due to a scarcity of eligible patients. Additionally, smaller facilities may refer patients to larger centers because of limited experience in treating the disease, rather than inadequate infrastructure. Notably, because naxitamab was not available in China, a number of patients from those sites were referred to Barcelona, Spain.

7. Methods, efficacy assessment – Did all sites perform bilateral marrow aspirates/biopsies or did some perform quadrilateral aspirates/biopsies? If any sites performed the latter, which were used for response assessment? Were antibodies targeting synaptophysin the only antibodies permitted as implied by the text?

REPLY: In accordance with the protocol, only bilateral marrow aspirates/biopsies were performed to align response assessment across patients. Only antibodies targeting synaptophysin were permitted.

8. Methods, efficacy assessment – The authors state, “If new soft-tissue lesions appeared post-baseline, these constituted PD.” They do not state whether there were any size criteria used to declare the presence of a new soft tissue lesion. Please clarify.

REPLY: Thank you for your comment. Soft tissue response was assessed as per the 2017 INRC, which advises using a combination of anatomic imaging and radionuclide scans to assess response in soft tissue and bone metastases. In accordance with the 2017 INRC, there were no specific size requirements for new soft tissue lesions to be categorized as PD; the appearance of any new soft tissue lesions that were MIBG- or FDG-PET-avid was categorized as PD. The published criteria, including size requirements, are addressed in the supplementary materials. (Park et al., 2017) The manuscript has since been updated to reflect that PD was determined according to the 2017 INRC criteria (page 6).

9. Results – The objective response rate is certainly of interest, but readers will also want to know how many patients had other categories of response. Please include the number of patients who had PD, SD, MD as relevant to the bone and bone marrow components of the INRC.

REPLY: Thank you for your comment. We have since added these important data to Table 2 (pages 40-43).

10. Results – The median duration of efficacy follow up was 6 months, and the minimum duration was one month. This is very short. The data cut-off was Dec 2021 and it is now 2024, which raises concerns. Were there additional data cut-offs in the time that has elapsed since Dec 2021? One wonders if the duration of response may have been estimable at one of those cut-offs. It is understood that the initiation of additional therapy such as vaccine therapy may have made this difficult, but readers will really want to know whether the effects of naxitamab are sustained or not, and the current data do not address that question well.

REPLY: We agree that additional follow-up data are important to assess the duration of response. There was only one preplanned interim analysis. The final study analysis will provide additional duration of response data.

11. Results – The same issue regarding timing of the PFS analysis (1-year PFS) arises. Since a fair amount of time has gone by since the time the cohort was enrolled and since Dec 2021, it seems reasonable to evaluate 2-year PFS in this cohort.

REPLY: We agree, and the final study analysis will provide these additional PFS data.

12. Results – Figure 2a provides some interesting information regarding response among patients with or without prior anti-GD2 antibody therapy, and the results are a bit surprising. Others have shown that patients who previously received (and may have responded) to anti-GD2 therapy tend to re-respond. However, that does not appear to be the case here as the response rate seems to be higher among patients who had not received prior anti-GD2 antibody therapy. This is not

mentioned in the text of the Results section but is interesting. This is mentioned in the discussion but it is not clear what the authors make of this.

REPLY:

Thank you for your comments. We agree that this is an interesting area of research, and the discussion in the manuscript has been expanded (page 16). It is worth underlining that Trial 201 lacks statistical power to perform subgroup analyses, which therefore requires caution in proposing causal explanations for observed subgroup differences. More precisely, additional research is required to clarify the basis of the differential response to naxitamab in complex and varied patients previously treated with another anti-GD2 mAb. The response to anti-GD2 therapy is a multi-determined outcome that may reflect unique PK/PD properties of anti-GD2 mAbs as well as patient and disease heterogeneity. For example, patients in Trial 201 who had received anti-GD2 therapy at any time prior to enrollment had a higher frequency of relapsed disease vs those with no prior anti-GD2 therapy. Relapsed disease is associated with a lower likelihood of response vs primary refractory disease (Moreno et al., 2017; Zhou et al., 2015). Interestingly, of the four patients who received an anti-GD2 mAb as part of their last treatment regimen prior to enrollment, two achieved CR.

We also wish to address the broader trend mentioned by the reviewer—namely that “*patients who previously received (and may have responded) to anti-GD2 therapy tend to re-respond.*” Of note are recent studies demonstrating that some patients who achieve remission with chemo-immunotherapy followed by a relapse respond when treated again with chemo-immunotherapy (Larrosa et al., ASCO, 2024). By contrast, Trial 201 investigated responses to naxitamab immunotherapy alone (with GM-CSF) for patients previously treated with relapse therapy who could not achieve remission prior to enrollment. This singular difference, as the reviewers rightly point out, limits direct comparisons of observed response rates in trials with distinctive designs and patient populations.

13. Discussion – The authors state that pain was intense but short lived, resolving within 15 minutes. However, data re duration of pain are not included in the Results section of the manuscript or in the supplementary materials. If duration of pain was measured, please add to the supplementary data section. If data regarding duration of pain were not collected, the sentence about this should be modified. The authors note that pain did not lead to treatment discontinuation for any patient. As noted below re Figure 1, it would be helpful to provide the reasons for treatment discontinuation.

REPLY: We have added tables summarizing median pain scores (Supplementary Table 7) and resolution of severe pain (Supplementary Table 8) to the supplementary material. In addition, in the Safety Results section, we state that the median pain score reverted to 0 at 15 minutes after infusion and remained at 0 prior to discharge for both scales (page 14). We have also clarified the adverse events (AEs) that led to treatment discontinuation in Figure 1 (page 32).

14. Discussion – The authors state, “The safety profile of naxitamab is manageable in the outpatient setting.” This requires at least a bit of qualification because sufficient infrastructure and personnel are in fact needed to support this (see comment 6 above). Also, none of the data in the rest of this paragraph are given in the Results section or in the supplementary materials. Please provide the data at least in the supplement.

REPLY: Thank you for your comments and suggestions. Site initiation meetings were conducted to ensure the sharing of best practices and the safe and effective administration of infusions and use of premedications, supportive therapies, and AE-management algorithms according to each site’s established processes and procedures. In the last paragraph of the Safety Results section (page 15), we state, “During the trial, 92% (1,144/1,237) of naxitamab infusions were administered in the outpatient setting, without the need for an overnight stay, and 99% (1,224/1,237) of infusions delivered the planned dose of naxitamab (3 mg/kg/day).” In the Discussion section (page 18), we have added references for expert consensus guidelines on pain mitigation and management strategies for naxitamab infusions and recently published experience with outpatient naxitamab administration at Atrium Health Levine Children’s Hospital. Results demonstrated that naxitamab can be administered and AEs effectively managed in outpatient settings within a dedicated multidisciplinary team supported by established processes and treatment algorithms.

15. Discussion – The concluding paragraph should be modified in light of the points made above.

Yes, thank you. We have updated our Discussion section and concluding paragraph to acknowledge the importance of an appropriately prepared multidisciplinary team (page 18).

Figures/tables

1. Figure 1 – What were the “other” reasons that patients were excluded in addition to not meeting eligibility criteria?

REPLY: Thank you for your question. We have revised Figure 1 (page 32) to list the “other” reasons that patients were excluded from eligibility. Six patients did not fulfill enrollment criteria due to requirements for disease status or time from start of induction therapy to trial enrollment (n=3); receipt of immunosuppressive agents (n=1); prior naxitamab treatment (n=1); or withdrawal of consent (n=1). One patient was considered better suited for other treatment at the investigator’s discretion.

2. Figure 1 – What were the specific adverse events that led to discontinuation of treatment?

REPLY: Thank you for your question. We have added the specific AEs that led to discontinuation of treatment in the footnotes of Figure 1 (page 32). These reasons included the following: anaphylaxis (n=2, both serious AEs [SAE] Grade 4); posterior reversible encephalopathy syndrome (n=1, SAE Grade 3); respiratory depression (n=1, SAE Grade 4); urticaria (n=1, SAE Grade 2); hypotension (n=1, non-serious treatment-emergent AEs Grade 2: occurred in the same patient who had respiratory depression leading to treatment discontinuation); myocarditis (n=1; SAE Grade 3) in a patient with a history of hypertrophic cardiomyopathy.

3. Table 1 – This study was conducted in the modern era, so stage should be given using INRG rather than INSS.

REPLY: Thank you for your comment. Reviewer 1 made a similar comment regarding staging criteria, addressed above, and now discussed in the Limitations section (pages 19 and 20). Briefly, this study was designed for regulatory purposes and to replicate the phase 1/2 trial (ISS 12-230) initiated in 2012 (Kushner et al., 2018), which utilized the INSS, and unfortunately, it is not possible to change the study design in the protocol at this time. Notably, INSS staging remains clinically relevant and is still currently used in clinical practice. Most patients in both the safety and efficacy populations had INSS stage 4 disease, which is equivalent to INRG stage M (Monclair et al., 2009). In addition, the Trial 201 inclusion criterion for HR disease at the time of diagnosis is as per INRG (Cohn et al., 2009).

4. Table 1 – Was INPC status formally assessed in all patients? This system is used routinely in North America however degree of differentiation is sometimes used in other countries to evaluate histology without the added elements of age and MKI included in INPC. Please confirm.

REPLY: International Neuroblastoma Pathology Classification (INPC) status was not assessed in this study.

5. Table 1 – The data re number of relapses are confusing. A quick look at the table suggests that the overwhelming majority of patients with relapsed disease were treated on this study in the first relapse setting. However footnote b says that 26 and 37 patients had relapsed at least once before enrollment. One would then think that these patients would not be considered to have been treated at time of first relapse. Please clarify. If these patients were indeed treated in first relapse, this should be added to the limitations section, especially in light of the data regarding the difference in response rates for patients who had received prior anti-GD2 antibody therapy vs those who had not. If the patients had had one prior relapse, then the heading in this row should be “Number of prior relapses” rather than “Number of relapses.”

REPLY: Relapsed patients were defined as those with an incomplete response to treatment for actively progressing or relapsed disease. Relapsed patients had therefore failed “salvage” therapy administered for the relapse or PD prior to enrollment. This definition is provided on page 6, and in the footnote of Table 1 (page 39). The header in Table 1 has been modified to clarify the number of prior relapses.

6. Table 2 – It will be easier for readers to think through this important table if all response are included and not just responses other than CR and PR.

REPLY: Thank you for your suggestion. We have added all responses to Table 2 (pages 40-42).

Reviewer #4 (Remarks to the Author): with expertise in neuroblastoma, therapy

The authors present the interim findings of study 201 which is a global, single arm, open label phase 2 study evaluating the combination of naxitamab and GMCSF for the treatment of patients with relapsed or refractory neuroblastoma with disease in the bone, bone marrow or both. I appreciate the authors presenting this data but feel that some changes are necessary. Below are the specifics of my review.

Abstract: Well written however, would consider changing the order presented in the results section to present the toxicity data followed by the response data. If this change is taken into consideration, recommend changing this order in the protocol “results” section as well. Additionally, the response data, as written in the results section of the abstract, is confusing. Would delete some of the response data in the abstract (such as the complete response information) and include more data on the toxicity. As an example, 62% of your patients had Grade 3 or 4 hypotension, but you do not include the number. The authors should include those numbers for the hypotension, pain and urticaria in the abstract.

REPLY: Thank you for your suggestions. This Abstract follows a format comparable to other published manuscripts evaluating similar therapies across efficacy, safety, and tolerability domains (page 4). We have done our best to accommodate your suggestions regarding inclusion of toxicity data in the Abstract while adhering to the journal-specified word limit. We have made updates to include the percentages of the most common naxitamab-related AEs.

Methods: The authors should state how a “bone lesion” was classified as both bone marrow and bone disease can be MIBG avid.

REPLY: Thank you for this clarification. The distinction between bone, BM, and bone + BM disease was determined by independent review. Briefly, bone vs BM disease was established by

assessing both the CS and BM pathology. For example, a negative BM biopsy/aspirate with CS >0 was classified as bone disease. We have added this explanation to the Methods (page 9).

Statistical Analysis: The authors should report why the patients with anti-drug antibody negative were excluded from the efficacy data. There is conflicting data about whether or not this effects response and thus these patients should not be excluded from the efficacy analysis.

REPLY:

Thank you for your question. Because this was the first study to formally assess efficacy and safety of naxitamab, patients who were positive for ADAs at baseline (as determined retrospectively) were excluded from the efficacy analysis to limit their potentially confounding effects. Importantly, patients whose ADA status changed over the course of the trial (ADA negative to ADA positive) were not excluded from the efficacy analysis.

We have completed a preliminary analysis of the potential impact of ADAs on the response to naxitamab therapy. The authors agreed that the PK/ADA results required thorough analysis and discussion and further agreed that these data required full exploration in a separate publication. Pending final guidance from the editor and reviewers, the authors would be amenable to including the PK/ADA in this manuscript and/or supplementary material.

Results: I would recommend presenting toxicity data prior to response data. Throughout the results, the authors used phrases like “CR rates were numerically higher” and others when describing data. Many of the presented findings were not statistically significant. Thus, those phrases should be removed. Instead the authors can just state the facts.

REPLY: Thank you for your comment. Because efficacy is the primary endpoint of the study, the authors opted to report efficacy data before safety data.

The manuscript has been edited to ensure that the data are clearly presented without unnecessary verbiage or generalizations. The manuscript clearly notes that the study was not powered for subgroup analysis, thereby preventing firm conclusions.

For the safety data, Naxitamab is not an easy medication to administer or for patients to receive. Thus, the authors need to provide more information as a supplemental table regarding the supportive care received by the patients. They currently have a figure with the supportive care possibilities. However, they supply no further information about what patients actually received. In the setting of some significant grade 3/4 toxicities, the community would benefit from more details. The authors also included duration (in hours) of certain Grade >3 toxicities. This is not something that is typically reported and should be removed. If the author is trying to make a point they can add that point to the Discussion Section.

REPLY: Thank you for your questions and comments. Supplementary Figure 3 (page 17) outlines the pre- and supportive medications available at bedside to manage acute AEs. Having readily available supportive therapies allows swift management of some of the most common and concerning side effects associated with naxitamab, such as anaphylaxis, nausea/vomiting, hypotension, bronchospasm, and pain. We have added this clarification to the Discussion (page 18). The manuscript further details the use of glucocorticoids in the management of bronchospasm and anaphylactic reactions (pages 15 and 18). Information on prior treatment and co-administered supportive therapies for the trial population was not available for this interim analysis, but it will be available for the final analysis after the data are fully vetted and analyzed.

Although the duration of certain Grade 3 to 4 toxicities may not be typically reported in other clinical trials, we believe that the duration of the most frequently reported and significant Grade 3-4 AEs provides valuable information for clinicians. Importantly, naxitamab is associated with severe infusion-related events and most events are manageable within a short period of time. For example, severe pain due to administration of anti-GD2 therapy can be significant enough to warrant hospital admission; however, treatment that successfully controls pain during an outpatient clinic visit can prevent hospital admission, unnecessary complications, and associated costs.

I think that it is very interesting that of the 50 SAEs, only 62% were considered related. Many of the side effects related to this therapy can also be related to supportive care and thus there must be overlap. The authors should include information regarding the SAEs that were “unrelated”. Again, in this section the authors use phrase like, “appeared to reduce” when describing the frequency of toxicities. Unless it is statistically significant, the authors should refrain from those sayings.

REPLY: We appreciate your thoughts regarding “unrelated SAEs” being potentially related to supportive care. Among the SAEs determined by investigators to be unrelated in this study, most would not be expected to be associated with naxitamab, but we acknowledge that any relationship to study treatment or supportive care is a subjective determination.

The table below lists the 19 SAEs considered “unrelated” to treatment (naxitamab or naxitamab + GM-CSF). Many of the listed AEs can be ruled out as related to supportive care.

	Patients
SAEs related to naxitamab or naxitamab+GM-CSF, n/N (%)	31/50* (62)
SAEs unrelated to naxitamab or naxitamab+GM-CSF, n/N (%)	19/50* (38)
Unrelated SAEs, n (%)	
Device-related infection	5 (10)
Febrile neutropenia	3 (6)
Pain	2 (4)
Pyrexia	1 (2)

Condition aggravated	1 (2)
Influenza	1 (2)
Generalized tonic-clonic seizure	1 (2)
Myelodysplastic syndrome	1 (2)
Mucoepidermoid carcinoma	1 (2)
Platelet count decreased	1 (2)
Hypomagnesemia	1 (2)
Pain in extremity	1 (2)

*Denominator represents the total number of events.

The manuscript has been edited to remove unnecessary verbiage and generalizations and clearly states that the study was not powered for subgroup analysis, thereby preventing firm conclusions.

Discussion: The discussion section is lacking discussion regarding other anti-GD2 therapies and how this treatment may compare. Further it is drawing conclusion (such as “response to naxitamab was similar across subgroups including MYCN amplification”) when these findings were not statistically significant AND the patient population had an unexpectedly low number of patients with MYCN amplification.

REPLY: Thank you for your comment. As noted above, direct comparisons between naxitamab and other anti-GD2 mAbs will require additional studies, including head-to-head trials. This need for additional studies is mentioned in the Discussion section (page 16).

Further, we now state that the study was not powered for subgroup analyses (limiting firm conclusions), while explicitly addressing the underrepresentation of MYCN amplification in the patient population as well (page 16).

The discussion section is lacking further discussion about the observed toxicities. It merely restates the results and does not elaborate. I think that it is prudent for the authors to discuss in further detail the 62% of patients with Grade 3 or greater hypotension and discuss some of the other results, not just restate them.

REPLY: Thank you for your comment. We have expanded the toxicity discussion to note that the frequency of Grade 3 pain and bronchospasm and Grade 3-4 infusion-related hypotension demonstrated reductions across treatment cycles, a change in trajectory of AEs that may inform mitigation strategies (pages 17 and 18). The high frequency of Grade 3 hypotension is also acknowledged in the manuscript. However, the vast majority of these events were manageable with the protocol guidance on AE management in place. This is supported by the following in the manuscript:

- 99% of infusions delivered the planned dose of naxitamab (3 mg/kg/day; page 18)
- A median infusion time of 37 minutes (maximum of 175 minutes; page 18)
- Only 1 patient was discontinued from naxitamab treatment due to hypotension (Supplementary Table 9; page 13)

Accordingly, we expand on appropriate AE management, including the use of consensus guidelines and recently published experience with outpatient naxitamab administration (page 18).

In general I think the tone of the manuscript and statements made throughout should be toned down and stick to the data and facts.

REPLY: Thank you for this suggestion. As noted above, we have updated the manuscript to address your specific concerns and to ensure that the data are presented without unnecessary verbiage or generalizations. Furthermore, we have clarified that the study was not powered for subgroup analysis, thereby preventing firm conclusions (page 16).

In addition, we have revised the manuscript to acknowledge that Trial 201 was conducted in a clearly defined population of patients with R/R neuroblastoma in the bone and/or BM compartments only. There is a need for further investigation and caution in interpreting the generalizability of findings across heterogeneous patients with HR neuroblastoma (page 16).

Figures/Tables:

Figure 1.

In figure 1, I am surprised that 10% of patients did not meet eligibility criteria and were excluded. This feels high. Additionally I have significant concerns that 10 patients did not have baseline disease evaluated per independent reviewer and another 5 did not have baseline bone marrow disease per independent review. That equals 20% of the analyzed patients included in the cohort. While I am pleased that the investigators removed them from the efficacy group, they did not discuss this in the manuscript and it makes me question other aspects of the data. This must be discussed in the body of the manuscript as the number is very high and technically that would mean that the patients should not have enrolled on the study.

REPLY: Thank you for your comments, and we understand your concern, which was also raised by another reviewer. We have edited Figure 1 (page 32) in response to your comments and in the Limitations section we acknowledge that a significant proportion of patients were excluded from the efficacy population (page 19). Please see response to reviewer 3 (3. Methods, patient population) for more details.

Figure 2.

Letter a figure should be removed and the data should be included in Table 2. Letter b data needs revision. It is impossible to see the overlying colors (relapsed/refractory and response). Further many of the lanes on the swimmers plots do not make sense and need to be double checked. Response End circles should just be removed as the overlay between that and response is difficult. As such, it is hard to see the progressive disease boxes.

REPLY: Thank you for your suggestions. Figure 2a has been removed and incorporated into Table 2. We have made changes to Figure 2 that are consistent with published swimmer plots while enhancing clarity. For example, the overlay was removed and the symbols were enlarged. In addition, we have confirmed that the data is accurately depicted in the figure.

Figure 3.

I would remove the OS Kaplan Meier curve as many patients went on to receive other therapy.

REPLY: Thank you for your suggestion. We acknowledge that many patients went on to receive other therapies, which confounds the OS results shown in the KM curve. In Figure 3, we also present the PFS Kaplan-Meier curve, for which the data were censored at the time that patients started a new therapy. While we agree that the OS Kaplan-Meier curve presents confounds, we believe it complements the data provided in the PFS Kaplan-Meier curve and provides an important perspective on the outcomes of high-risk patients, who often participate in clinical trials over the course of care. Nevertheless, we recognize the risk for misinterpretation, and defer to the reviewer's judgment on whether the OS Kaplan-Meier curve should be removed. The definitions of both outcome measures and the data censoring approaches have been clearly defined in the legend for Figure 3.

Table 1. MYCN is under-represented in this R/R patient population at only 14%. Thus, the authors should discuss this in the body of the manuscript.

REPLY: Thank you for your comment. We acknowledge the low percentage of *MYCN* amplification in this study population and have added this to the discussion (page 16).

Table 3.

When reviewing the supplementary table and this table I think that a few changes should be made. Recommend including the Grade 4 anaphylaxis and the Grade 4 respiratory depression in the supplementary table in Table 3 instead.

REPLY: Thank you for this suggestion. Table 3 reflects all related AEs (irrespective of grade) occurring in $\geq 10\%$ of patients, as well as associated Grade 3 to 4 AEs (i.e., Grade 3 and 4 AEs reported within the same preferred terms). The events of Grade 4 anaphylaxis and Grade 4 respiratory depression did not occur in $\geq 10\%$ of patients in Table 3, but they are shown in

Supplementary Table 9, Related AEs that led to treatment discontinuation in Trial 201.

References

- Cheung IY, Kushner BH, Modak S, Basu EM, Roberts SS, Cheung NV. Phase I trial of anti-GD2 monoclonal antibody hu3F8 plus GM-CSF: impact of body weight, immunogenicity and anti-GD2 response on pharmacokinetics and survival. *Oncoimmunology*. 2017;6(11):e1358331.
- Cohn SL, Pearson AD, London WB, et al. The International Neuroblastoma Risk Group (INRG) classification system: an INRG Task Force report. *J Clin Oncol*. 2009;27(2):289-297.
- DuBois SG, Macy ME, Henderson TO. High-risk and relapsed neuroblastoma: toward more cures and better outcomes. *Am Soc Clin Oncol Educ Book*. 2022;42:1-13.
- Flaadt T, Ladenstein RL, Ebinger M, et al. Anti-GD2 antibody dinutuximab beta and low-dose interleukin 2 after haploidentical stem-cell transplantation in patients with relapsed neuroblastoma: a multicenter, phase I/II trial. *J Clin Oncol*. 2023;41(17):3135-3148.
- Irwin MS, Naranjo A, Zhang FF, et al. Revised neuroblastoma risk classification system: a report from the Children's Oncology Group. *J Clin Oncol*. 2021;39(29):3229-3241.
- Karnofsky DA, Burchenal JH. The clinical evaluation of chemotherapeutic agents in cancer. In: MacLeod CM, ed. *Evaluation of Chemotherapeutic Agents*. New York, NY: Columbia University Press; 1949:191-205.
- Kushner BH, Cheung IY, Modak S, Basu EM, Roberts SS, Cheung NK. Humanized 3F8 anti-GD2 monoclonal antibody dosing with granulocyte-macrophage colony-stimulating factor in patients with resistant neuroblastoma: a phase 1 clinical trial. *JAMA Oncol*. 2018;4(12):1729-1735.
- Lansky SB, List MA, Lansky LL, Ritter-Sterr C, Miller DR. The measurement of performance in childhood cancer patients. *Cancer*. 1987;60(7):1651-1656.
- Larrosa C, Simao M, Muñoz JP, et al. Naxitamab chemoimmunotherapy regimens other than with irinotecan/temozolomide for patients with relapsed/refractory high-risk neuroblastoma [poster]. Presented at American Society of Clinical Oncology Annual Meeting, May 31-June 4, 2024, Chicago, IL.
- Lode HN, Ehlert K, Huber S, et al. Long-term, continuous infusion of single-agent dinutuximab beta for relapsed/refractory neuroblastoma: an open-label, single-arm, phase 2 study. *Br J Cancer*. 2023;129(11):1780-1786.
- Mody R, Naranjo A, Van Ryn C, et al. Irinotecan-temozolomide with temsirolimus or dinutuximab in children with refractory or relapsed neuroblastoma (COG ANBL1221): an open-label, randomised, phase 2 trial. *Lancet Oncol*. 2017;18(7):946-957.
- Mody R, Yu AL, Naranjo A, et al. Irinotecan, temozolomide, and dinutuximab with GM-CSF in children with refractory or relapsed neuroblastoma: a report from the Children's Oncology Group. *J Clin Oncol*. 2020;38(19):2160-2169.
- Monclair T, Brodeur GM, Ambros PF, et al. The International Neuroblastoma Risk Group (INRG) staging system: an INRG Task Force report. *J Clin Oncol*. 2009;27(2):298-303.

- Moreno L, Rubie H, Varo A, et al. Outcome of children with relapsed or refractory neuroblastoma: a meta-analysis of ITCC/SIOPEN European phase II clinical trials. *Pediatr Blood Cancer*. 2017;64(1):25-31.
- Moreno L, Weston R, Owens C, et al. Bevacizumab, irinotecan, or topotecan added to temozolomide for children with relapsed and refractory neuroblastoma: results of the ITCC-SIOPEN BEACON-neuroblastoma trial. *J Clin Oncol*. 2024;42(10):1135-1145.
- Park JR, Bagatell R, Cohn SL, et al. Revisions to the International Neuroblastoma Response Criteria: a consensus statement From the National Cancer Institute Clinical Trials Planning Meeting. *J Clin Oncol*. 2017;35(22):2580-2587.
- Trovillion EM, Michael M, Jordan CC, et al. Guidelines for outpatient administration of naxitamab: Experience from Atrium Health Levine Children's Hospital. *Cancer Med*. 2024;13(3):e7045.
- Zhou MJ, Doral MY, DuBois SG, Villablanca JG, Yanik GA, Matthay KK. Different outcomes for relapsed versus refractory neuroblastoma after therapy with (131)I-metaiodobenzylguanidine ((131)I-MIBG). *Eur J Cancer*. 2015;51(16):2465-2472.

REVIEWER COMMENTS

Reviewer #1 (Remarks to the Author): The authors have satisfactorily responded to the review.

RESPONSE: Thank you for reviewing the manuscript and for your helpful feedback.

Reviewer #2 (Remarks to the Author):

The authors responded well to the reviewers' comments. I think it is very important for the field that the results of this trial are published. Even though I still feel that the results are overestimated due to a highly selected cohort of patients, I think the manuscript is ready for publication, and I have no further comments.

RESPONSE: Thank you for your kind comments and for acknowledging the importance of the Trial 201 results to the clinical community. We agree that these findings are highly relevant and appreciate your thorough review and helpful feedback. As discussed below in some detail, we agree that this carefully selected study population does not include all patients with refractory and relapsed disease as now defined in the literature. We therefore present the results in the Discussion and Limitations sections within a community-based context spanning varied subpopulations with distinct treatment needs and historical outcomes, and tumor and disease burdens.

Reviewer #4 (Remarks to the Author):

Comment 1

The abstract, as written, is impossible to follow. For example, it reads “among 26 responders” followed up by (CR+PR; n=52)...this does not make sense unless you read the manuscript and realize that 22 patients were excluded from the response analysis. In the abstract, the authors need to do a better job describing the results. It is too confusing as written. I believe a sentence in the abstract stating that 22 patients were excluded from the efficacy analysis will make this more clear.

RESPONSE: Thank you for your comments and suggestion. We have edited the abstract to clarify that there were 26 responders in the efficacy population with due sensitivity to the word count and can expand in greater detail, as needed.

Comment 2

While the study evaluates an important drug and there are encouraging results for a specific patient population, I feel that the trial and interim analysis as presented does not meet the scientific rigor and requirements expected from studies published in Nature Communications. are too many risks of potential biasing of data including the enrollment criteria and definition and number of patients excluded from the efficacy data. Additionally, the lack of transparency around some of the toxicity data leads to questions. I appreciate the revisions and feel that the revised manuscript is improved. However, I think the number of inherent flaws in the clinical trial design and included data are concerning unless further addressed.

RESPONSE: Thank you for your comments and helpful feedback. We agree that the results are encouraging for this patient population and appreciate your concerns about the number of patients excluded from the efficacy population and the potential risk of bias. We hope that providing additional context will satisfactorily address your comments.

We now state in the Limitations section that overall, a significant proportion of patients (n=22) were excluded from the efficacy population. As noted previously, 10 patients were excluded from the efficacy population due to lack of evaluable disease at baseline based on an independent analysis of the baseline scans (¹²³MIBG and CT/MRI scans) and bone marrow histopathology. Trial 201 was explicitly designed to evaluate ORR in patients with residual disease in the bone/BM, so evaluable disease at baseline is a threshold requirement to gauge treatment response. The protocol therefore stipulates that enrolled patients without evaluable baseline disease were to be included in the safety analysis set (SAF)—thereby contributing to the assessment of the naxitamab safety profile—and excluded from the efficacy population. Importantly, this pre-specified exclusion resulted in a higher overall disease burden among the efficacy population.

In accordance with the Trial 201 inclusion criteria, an additional 5 patients were excluded because their baseline disease was insufficient as per independent review (BM infiltration of <5% with no bone disease). This, too, resulted in a higher disease burden among those in the efficacy population. Additional details are provided in our response to Comment 5 below. Briefly, exclusion of these 15 patients aligns with the 2017 International Neuroblastoma Response Criteria (INRC) and published consensus guidelines for early-phase clinical trial eligibility and response evaluation criteria for R/R neuroblastoma^{1,2}.

In the Methods section, we now expand on the process by which these 15 patients were evaluated over the course of the trial. The patients were enrolled at the investigator's discretion, and an independent, central committee subsequently reviewed each patient's baseline data set based on criteria pre-specified in the protocol. Independent review is a regulatory requirement to ensure unbiased and consistent response assessments. Since the patients did not fulfill protocol-specified criteria for Trial 201, they were retrospectively excluded from the efficacy analysis. A prospective and centralized review by this independent committee would have significantly delayed enrollment and treatment with naxitamab, presenting both practical and ethical challenges.

This is the first trial rigorously investigating the efficacy and safety of naxitamab in well-defined patients with HR neuroblastoma. Eligibility criteria were therefore established to minimize potential confounds. Given the uncertainty surrounding the relationship between anti-drug antibodies (ADAs) and response, the protocol pre-specifies the exclusion of patients with baseline ADAs from the efficacy analysis and their inclusion in the overall safety population. It is worth noting that patients who developed ADAs during treatment were not excluded from the efficacy analysis. At the time of trial design, the ADA assay had not been validated, so the samples were collected from trial participants and assessed retrospectively. Of the four excluded patients with positive ADA status, one achieved a CR (responder) and another had SD (non-responder); their exclusion therefore had no impact on the ORR. The two remaining patients with ADAs were enrolled just prior to the data cutoff, and their response data had not been independently evaluated for this interim analysis, further warranting their pre-specified exclusion.

Overall, the low incidence of baseline ADA positivity (5.4%, 4/74) in the study population suggests a low risk of bias and supports the application of these results to community-based patients with R/R HR neuroblastoma and residual disease in the bone and/or BM only. We implicitly defer to the editor's and reviewers' discretion on whether these and related data on the ADA status should be included in the manuscript and supplementary materials.

Two patients were excluded from the efficacy analysis because they received radiotherapy to target metastatic site during trial (with the potential to bias the efficacy of naxitamab), and one patient was excluded because the first response assessment occurred after the data cutoff date for the interim analysis.

In conclusion, we believe the protocol's criteria—pre-specified to define a target population—and the peer-reviewed literature provide a cogent rationale for excluding the 22 patients from the efficacy analysis. Lastly, and perhaps most importantly, we are committed to transparent data sharing and are eager to address any specific questions or concerns regarding the toxicity data.

Comment 3

In the limitations section, more time must be focused on the inclusion criteria of this study in that it does not represent many of the patients one thinks of when considering a relapsed or refractory NB patient. The average reader, lacking neuroblastoma expertise, may not be able to distinguish that the studied patient population is very specific and represents a small number of R/R patients. For example, it is worth noting in the manuscript that this patient population likely represents a skewed patient population from prior reported clinical trials. This is evidenced in the baseline Lansky and Karnofsky scores being 100% and 96%. This likely is a reflection of minimal disease present at baseline and must be addressed in the limitations section.

RESPONSE: Thank you for your helpful feedback. To further address your concerns regarding the study population, and to minimize the risk of misinterpretation, we have made additional revisions in the manuscript.

First, we note in the Introduction section that HR neuroblastoma is a complex disease characterized by heterogeneous tumors and patients as well as differential responses within and between patients over the course of treatment. We further note in the Methods section that this variability required careful selection of the Trial 201 study population to determine the safety and efficacy of naxitamab when administered as a single-agent therapy (i.e., not combined with chemotherapy). Elaborating further in the Discussion section, we cite numerous studies characterizing the neuroblastoma tumor microenvironment that may limit the efficacy of single-agent antibody therapy³⁻⁵. We also summarize seminal work by Mody et al. on dinutuximab in combination with irinotecan and temozolomide (DIT) in patients with relapsed or refractory HR neuroblastoma⁶. Briefly, this study showed limited efficacy for DIT in patients with soft tissue lesions (STLs, or measurable disease, 21.6% overall response) vs those with evaluable disease in bone/BM (OR, 87.5%). Furthermore, among those with best response of SD, 95.5% had measurable disease, and 68.1% had SD because of lack of response in soft tissue. Current evidence therefore supports chemotherapy alone or in combination with anti-GD2 mAbs as the standard of care for the treatment of metastatic STLs.

Accordingly, we highlight in the Introduction and Discussion sections that Trial 201 was explicitly designed to investigate naxitamab monotherapy (with GM-CSF) in patients with R/R HR neuroblastoma and residual disease in bone and BM, objectives that required exclusion of those with STLs. Relatedly, the trial also excluded patients with actively relapsing disease and the most aggressive forms of HR neuroblastoma (including what one reviewer described as galloping HR neuroblastoma), for whom more intensive therapies than single-agent naxitamab therapy are also indicated.

We acknowledge in the Limitations sections that refractory HR neuroblastoma has varied definitions. As part of our broader commitment to elucidating the role of anti-GD2 monotherapy in select subpopulations, this study defines refractory disease as an incomplete end-of-induction (EOI) response with residual disease that is limited to bone and BM. While this definition is inclusive of some though not all definitions in the published literature, it nonetheless builds on recent evidence demonstrating the dual importance of targeting residual bone/BM disease and achieving a complete response prior to consolidation⁷⁻¹⁰. In the Introduction section, we further note that previous studies have not evaluated treatment options for the bone and BM compartments in isolation, despite their widely accepted role as a metastatic niche for chemoresistant neuroblastoma cells. In sum, the study defines the eligibility criteria with a high level of precision to permit the investigation of naxitamab monotherapy in relatively homogeneous subpopulations, without the confounding effects of concurrent chemotherapy.

We acknowledge in the Introduction, Discussion, and Limitations section that this study does not include all patients with R/R disease as now defined in the literature, defining a gap that must be explored in future research. We are, moreover, aligned with your comments, and those shared by the other reviewers as well, that this targeted study population inherently limits the broader application of the results. This, of course, holds true for any trial designed to prospectively evaluate two subpopulations—in this case, those with R/R disease—to the exclusion of others. We therefore present the results in the Discussion and Limitations sections within a community-based context spanning varied subpopulations with distinct tumors and disease burdens, treatment needs and historical outcomes.

Thank you for your thoughtful observation regarding the Lansky and Karnofsky scores. The study population in Trial 201 comprises a well-defined cohort of patients with relapsed/refractory HR neuroblastoma, with the protocol excluding patients with Karnofsky/Lansky scores of <50%. Notably, 58% of the efficacy population had a baseline Curie score (CS) of ≥ 3 , and of the 66 patients with performance scores ≥ 90 , 28 (44%) also had a CS of ≥ 3 . There is limited research evaluating the relationship between performance scores and disease burden, and you have identified an important issue that requires focused investigation. Our clinical experience supports your observation that performance scores for patients with actively progressing disease would on average likely be lower compared with those included in Trial 201. We have adjusted the Limitations section accordingly.

Comment 4

While the language in the discussion section has been toned down, I still feel that the results are overstated and that the authors did not adequately address the limitations and potential bias of the results. With this patient population, it is very difficult to make sweeping generalizations about comparison to other studies.

RESPONSE: Thank you for your feedback. In the last submitted version of the manuscript, we included a Limitations section to address concerns related to potential biases and limitations of our results. As noted above, we agree with your assessment that it is challenging and indeed inadvisable to make broad comparisons to other studies. HR neuroblastoma is characterized by a diverse patient population with variable clinical presentations, treatment needs, and outcomes^{3,4}. Recent studies of inter-tumor and intra-tumor differences and the role of a complex tumor microenvironment further demonstrate disease heterogeneity as a cardinal feature of HR neuroblastoma and an increasingly important focus of clinical research³⁻⁵. Despite progress, there remains a significant gap in the literature on the current and emerging role of targeted therapy—in particular anti-GD2 monoclonal antibodies—in patients with select clinical and histopathological characteristics^{11,12}.

This manuscript addresses this gap in several areas. First, it presents clinically meaningful improvements in well-defined patients with residual disease in the bone and BM only. Despite research demonstrating that residual disease in the bone/BM is associated with significant chemoresistance and is predictive of poor outcomes^{13,14}, studies have not evaluated the safety and efficacy of anti-GD2 therapy in this specific patient population. Second, the manuscript evaluates patients with refractory HR neuroblastoma and residual disease in bone/BM, defined as an incomplete response (PR, MR, or SD per the 2017 International Neuroblastoma Response Criteria [INRC]) to frontline therapy. This definition is supported by seminal studies showing the importance of achieving a complete response and minimizing disease burden prior to consolidation⁷⁻¹⁰. That there is no consensus definition of refractory HR neuroblastoma partly reflects the singular challenges of investigating a complex, rare, and heterogeneous pediatric malignancy. We believe this manuscript—with its focus on targeted subpopulations—offers an evidence-based contribution to the discussion.

The manuscript further presents data supporting the treatment of patients with relapsed disease, defined as an incomplete response to treatment for actively progressing or relapsed disease, also with residual disease limited to the bone/BM, as per the 2017 INRC. Relapsed disease is associated with a dismal prognosis, further underscoring the rationale for focused investigation of this subpopulation and the need for additional effective treatments.

Overall, the manuscript presents the interim findings for naxitamab monotherapy (+GM-CSF) in vulnerable and carefully defined patients with HR neuroblastoma, part of an emerging effort to tailor targeted immunotherapies based on an improved clinical and biological understanding of the disease. Importantly, we do not extrapolate the results from Trial 201 to other study populations. Moreover, we acknowledge in the Discussion and Limitations sections the limited generalizability of our results across a heterogeneous patient population, while underlining the need for additional studies, including in patients with soft tissue and progressive disease.

We appreciate your insights and would welcome any specific suggestions you have regarding additional limitations or biases that require additional clarification in the manuscript.

Comment 5

I still have significant concerns about the patients that were excluded (15 of 74 patients) from the efficacy data. The information provided in the authors response has heightened my concerns with bias and I think this needs to be further addressed.

RESPONSE: Thank you for your comments and questions. As per the study protocol and consistent with best practices for clinical trials, the outcome measures required evaluable disease at baseline. As noted above, in Trial 201, 10 patients were excluded from the efficacy population for lack of evaluable disease at baseline per independent review, a core regulatory requirement for evaluating treatment response. In addition, as per pre-specified inclusion criteria, 5 patients were excluded for insufficient baseline disease, which was defined as BM infiltration <5% with no bone disease, determined independently.

Eligibility and response evaluation criteria for early-phase clinical trials in refractory, relapsed, or progressive neuroblastoma have been previously established². According to the consensus statement, inclusion in the efficacy analysis of the primary endpoints (ORR) requires patients to have at least one tumor site that is evaluable for response assessment; therefore, excluding 10 patients due to a lack of baseline disease is supported. In addition, bone marrow disease is considered evaluable only if there is >5% tumor involvement in any single sample from bilateral aspirates and biopsies^{1,2}. Consequently, the exclusion of 5 patients with <5% bone marrow infiltration and no bone disease is also supported. Similar criteria are currently used in other early-phase neuroblastoma clinical trials. For example, in the ongoing phase 2 Children's Oncology Group study, ANBL1821 (Clinicaltrials.gov identifier: NCT03794349), investigating irinotecan/temozolomide/dinutuximab with or without eflornithine in children with relapsed, refractory, or progressive neuroblastoma, the inclusion criteria specify that patients with bone marrow disease are eligible only if there is >5% disease involvement (i.e., documented neuroblastoma cells) in at least one sample from bilateral bone marrow biopsies.

Lastly, it is noteworthy that patients in Trial 201 (n=3) were classified as non-responders in the pre-specified interim analysis if they had evaluable disease at baseline but were subsequently determined to have non-evaluable disease for any reason at any point after baseline.

References

1. Park JR, et al. Revisions to the International Neuroblastoma Response Criteria: a consensus statement From the National Cancer Institute Clinical Trials Planning Meeting. *J Clin Oncol* **35**, 2580-2587 (2017).
2. Park JR, et al. Early-phase clinical trial eligibility and response evaluation criteria for refractory, relapsed, or progressive neuroblastoma: a consensus statement from the National Cancer Institute Clinical Trials Planning Meeting. *Cancer* **128**, 3775-3783 (2022).
3. Gomez RL, Ibragimova S, Ramachandran R, Philpott A, Ali FR. Tumoral heterogeneity in neuroblastoma. *Biochim Biophys Acta Rev Cancer* **1877**, 188805 (2022).
4. Polychronopoulos PA, Bedoya-Reina OC, Johnsen JI. The neuroblastoma microenvironment, heterogeneity and immunotherapeutic approaches. *Cancers (Basel)* **16**, 1863 (2024).
5. Louault K, De Clerck YA, Janoueix-Lerosey I. The neuroblastoma tumor microenvironment: from an in-depth characterization towards novel therapies. *EJC Paediatr Oncol* **3**, 100161 (2024).
6. Mody R, et al. Irinotecan, temozolomide, and dinutuximab with GM-CSF in children with refractory or relapsed neuroblastoma: a report from the Children's Oncology Group. *J Clin Oncol* **38**, 2160-2169 (2020).
7. Streby KA, et al. Impact of diagnostic and end-of-induction Curie scores with tandem high-dose chemotherapy and autologous transplants for metastatic high-risk neuroblastoma: a report from the Children's Oncology Group. *Pediatr Blood Cancer*, e30418 (2023).
8. Pinto N, et al. Predictors of differential response to induction therapy in high-risk neuroblastoma: a report from the Children's Oncology Group (COG). *Eur J Cancer* **112**, 66-79 (2019).
9. Yanik GA, et al. Semiquantitative mIBG scoring as a prognostic indicator in patients with stage 4 neuroblastoma: a report from the Children's oncology group. *J Nucl Med* **54**, 541-548 (2013).
10. Yanik GA, et al. Validation of postinduction Curie scores in high-risk neuroblastoma: a children's oncology group and SIOPEN group report on SIOPEN/HR-NBL1. *J Nucl Med* **59**, 502-508 (2018).
11. Han JZR, Hastings JF, Phimmachanh M, Fey D, Kolch W, Croucher DR. Personalized medicine for neuroblastoma: moving from static genotypes to dynamic simulations of drug response. *J Pers Med* **11**, 395 (2021).
12. Lundberg KI, Treis D, Johnsen JI. Neuroblastoma heterogeneity, plasticity, and emerging therapies. *Curr Oncol Rep* **24**, 1053-1062 (2022).
13. Lin KS, et al. Minimal residual disease in high-risk neuroblastoma shows a dynamic and disease burden-dependent correlation between bone marrow and peripheral blood. *Transl Oncol* **14**, 101019 (2021).
14. Yang R, Zheng S, Dong R. Circulating tumor cells in neuroblastoma: current status and future perspectives. *Cancer Med* **12**, 7-19 (2023).